# Visual Design Cues Impacting Food Choice: A Review and Future Research Agenda

**DOI:** 10.3390/foods9101495

**Published:** 2020-10-19

**Authors:** Iris Vermeir, Gudrun Roose

**Affiliations:** 1BE4LIFE, Department of Economics and Business Administration, Ghent University, Tweekerkenstraat 2, 9000 Ghent, Belgium; g.roose@ieseg.fr; 2IESEG, School of Management, 59000 Lille, France

**Keywords:** visual cue, food choice, food perception, attitudes, behaviour, psychological processes, taste, healthy, sustainable, quality, point of purchase

## Abstract

This review aims to tackle the challenge of understanding how visual design cues can affect behavioural outcomes in a food context. The review answers two key questions: (1) What are the effects of the most important visual design cues on behavioural outcomes and how can they be explained? (2) What are the research gaps in this area? We start from a comprehensive taxonomy of visual design cues delineating the most important visual design cues. Next, we evaluate the extant research based on a structured, narrative literature review on visual design cues in the food domain. We differentiate between object processed and spatially processed visual design cues in food choice contexts and show how they affect behavioural outcomes through a range of psychological processes (attention, affective-, cognitive- and motivational reactions, food perceptions and attitudes). We end with recommendations which take into account the current food store context, the state-of-art in measuring psychological processes and behavioural outcomes and the specific food-, person- and context-related moderators. This review offers guidance for research to untangle the complexity of the effect of visual design cues in a food choice context.

## 1. Introduction

Every day, people are faced with a multitude of food choices ranging from choosing a particular food product (for example a healthy or sustainable option) to choosing how much food to buy. People often choose which food product they want to buy within a few seconds [1,2,3,4]. Food choices are often made on the spot, at the (online or offline) point where the food is offered to them [5,6].

In a retail context, both online and offline, much of the information visitors use to make these choices is predominantly visual in nature. When we enter a physical (food) store or visit an online (food) store, we are exposed to a multitude of visual cues or external, physical characteristics [7] of the product, packaging or environment that create the appearance of a product. Visual cues entail graphical elements, like advertising colours or type font and structural elements like package size and shape and materiality [8]. Also spatial elements like the design of in-store/online displays and advertisements, product presentation and logo or label orientation and positioning both online and offline are visual cues [9].

Visual cues are one of the most significant communication tools of modern society [7,8,10]. From a marketing perspective, visual cues are used to draw attention [11,12,13], provide sensory pleasure and stimulation and communicate about a product or brand (image) [14]. For an individual, visuals can be aesthetically pleasurable [15], provide information [16] and help us make sense of what we see [17]. The first taste is through the eyes [18]: people derive food expectations from visual properties of food [14,19,20,21] on for example ingredients but also on volume, quality, taste [22] and health [23,24,25,26,27] (namely, cue utilization process, [28]). Before consumption, especially when people have no experience with a product, they must rely on visual cues to evaluate food [8,29,30]. Visual cues then deliver a salient, vivid, and strong input for choices between different food products and brands [31]. In most choice contexts, people often only use visual cues to make choices when these cues are readily available and salient [32,33] so it is imperative that visual cues catch peoples’ eye. In a food choice context, choices are often made quite automatically without much thought based on cues that attract attention in the choice context [34].

The goal of this paper is to bring together research on the effect of visual cues on behavioural outcomes in a food context and advance future research by providing interesting research gaps in this area. We focus on ‘whether’ and ‘why’ specific visual cues affect behavioural outcomes. We describe previous research showing ‘whether’ specific visual cues influence behavioural outcomes (e.g., food choices) and discuss ‘why’ these specific visual cues can affect behavioural outcomes at the choice context through psychological processes like attention, cognitive, affective and motivational reactions, product perceptions and attitudes [9,13,16,19,35,36,37,38,39,40,41,42,43,44] (see Table 1).

From now on, we refer to visual cues as ‘visual design cues’ to emphasize that we concentrate on visual cues that are related to the ‘look’, ‘appearance’ or ‘form’ of the visual cue rather than the content of the visual cue. Visual design cues are for example the product or logo colour, package shape, typeface of words on packaging, or food presentation rather than the content of an in-store advertising appeal (e.g., the presentation of a family having diner) or the content of a logo (e.g., the logo of a soft drink company representing a sunset). The current review does not incorporate visual cues in out-of-store advertising but rather focusses on the effect of visual design cues on the moment the actual choice is made. We also do not concentrate on visual processing (i.e., the reception and automatic representation of stimuli in the brain as influenced by individuals’ internal states; [7]) or visual comprehension (i.e., the categorizations and holistic evaluations that individuals make regarding perceived stimuli; [7]). For reviews on visual processing and visual comprehension we refer respectively to [45,46,47] and [48,49].

We started from a comprehensive taxonomy of visual design cues based on recent work of Sample et al. [7] and Adaval et al. [50]. We then critically reviewed the literature on the effect of visual design cues on behavioural outcomes and psychological processes. Our intended contribution is to answer two key questions: (1) What are the effects of the most important visual cues on behavioural outcomes and how can they be explained? (2) What are the research gaps in this area? The current work aims to contribute by structuring the highly fragmented and context specific work on the effect of visual design cues [49] in the food domain.

The current review differs from previous reviews in several respects. First, we combine a top-down approach with a bottom-up literature review. That is, we start from a comprehensive taxonomy of visual design cues that delineates the most important visual design cues in a point of purchase context. Next, we evaluate the extant research based on a structured, narrative literature review on visual design cues in the food domain. We also added research from a non-food context and from non-food domains in case this research and these theories could be applied to behavioural outcomes in a food context. Based on this literature review, we identify interesting research questions. Most other reviews limit their literature review to a specific context (for example, advertising) or domain (for example, food). Because we bring together theories of different domains and contexts, this review is well positioned to uncover gaps in food choice research. In addition, we not only focus on the plausible behavioural effects of visual design cues in a food choice context but also take into account ‘why’ these visual design cues could have an effect (i.e., underlying psychological processes). Most other reviews limited their literature review to studying effects of visual design cues in specific contexts and not describing why these effects could occur.

In the remainder of the document, we start off with discussing which typologies of visual design cues exist and shortly explain the typologies of Sample et al. [7] and Adaval et al. [50]. Then, we provide a non-exhaustive review of visual design cues that affect behavioural outcomes and psychological processes in a food choice context. A list of specific behavioural outcomes and psychological processes that we discuss can be found in Table 2. We provide examples from research showing the effect of visual design cues on behavioural outcomes and psychological processes and list future research questions regarding visual design cues in a food choice context. Following previous research, we differentiate between two broad categories of psychological processes that visual design cues may affect: (1) product perceptions and attitudes and (2) other psychological processes (e.g., attentional, affective, cognitive and motivational reactions) [39] (see Figure 1). As shown in Figure 1, visual design cues can affect behavioural outcomes through product perceptions, attitudes and other psychological processes. Product perceptions and attitudes can also mediate the relationship between other psychological process variables and behavioural outcomes.

## 2. Taxonomy of Visual Design Cues Impacting People’ Food Decisions

Typologies or taxonomies of visual design cues are discussed in a variety of research domains: neuroscience, psychology, computer science (information visualization), marketing, (video) advertising, and online visual merchandising (VDM) [7,16,50,51,52,53,54,55]. Although differently categorized or defined, these typologies often present more or less the same type of visual design cues. Common cues are: colour, shape, location (e.g., camera angle), aesthetic cues (e.g., visual complexity, balance), etc.

In this paper we use the typologies of Adaval et al. [50] and Sample et al. [7] to structure the literature. Other existing typologies are definitely valuable, but more limited or specific in scope. Tory and Moller [51], for example, concentrate on the classification of visual algorithms instead of specific cues; Phillips and McQuarrie [52] solely focus on visual rhetorical figures; Larsen et al. [53] focus on video advertising; Ha et al. [54] merely investigate VDM and Rossiter et al. [55] focus only on advertising. We did not adopt the typology of Raghubir [16] either as this typology includes both visual *content* related cues and visual *design* related cues. Nevertheless, the visual *design* cues discussed by Raghubir [16] coincide more or less with those discussed by Adaval et al. [50] and Sample et al. [7].

We focus on the typologies of Adaval et al. [50] and Sample et al. [7] and link them to each other. We selected these typologies for the following reasons: (1) they are focusing on purely the ‘design’ of the visual cue and not on the content represented by the visual cue; (2) these typologies represent visual design cues which matter at the point of purchase and apply to all kinds of marketing media and do not only focus on, for example, advertising (e.g., static images or video) and (3) these typologies allow us to combine a top-down conceptualization of visual design cues (as used by Adaval et al. [47]) with a bottom-up identification of visual design cues (as used by Sample et al. [6]) into one framework.

Adaval et al. [50] builds a taxonomy of visual design cues using a top-down approach. They categorize visual design cues by how we process them: (1) object processing and (2) spatial processing [50]. Object processing refers to the identification and recognition of cues in the environment shaped by existing concepts and associations in our memory [50] (p. 50). In short, object processing is about ‘what’ we process. Spatial processing, in contrast, is about the visual perception of location of objects, their movement, and spatial relation between the object and the self. As such, spatial processing refers to ‘where’ we process.

Sample et al. [7] embrace a bottom-up approach for defining a typology of visual design cues by identifying and analysing the individual components and facets (sub-components) of visual perception. Although Sample et al. [7] do not make a distinction between ‘what’ we process and ‘where’ we process, their five main components can be categorized along the dichotomy made by Adaval et al. [50]. Table 2 provides an overview of the typology of Adaval [50] and Sample et al. [7] and how these typologies more or less align. Within object processed cues we discern 5 subcategories relevant for food choices at the point of purchase: (1) colour, (2) shape, (3) aesthetic cues, (4) materiality and (5) text and picture combinations. Within spatial processed cues we distinguish three subcategories: (1) location, (2) movement and (3) spatial relation between object and self.

All of these visual design cues are more or less manifested in food research. Table 3 provides examples of visual design cues in a food context.

## 3. Method

We used a desk research method to build a framework of visual design cues to organize the relevant literature. We consecutively performed a broad structured literature search, screened papers on relevance and quality and performed a second structured literature search to find out the effects of the most important visual design cues on food choices and research gaps in this area.

To select the literature, we conducted a structured literature search in Web of Science combining specific keywords indicating the visual perception character (e.g., visual perception, visual cue) with the marketing aspect (e.g., product perception, choice) and the food aspect (e.g., food). This resulted in 6449 papers. We refined our search to particular domains (food sciences, behavioural sciences, business, psychology, economics, graphic arts, architecture, engineering and agricultural economics) and papers published between 2010 and 2020. Within the frame of selected papers published between 2010 and 2015, we only selected those that were cited three or more times (indicating the paper’s relevance) resulting in 866 papers.

These 866 papers were screened on quality and relevance which were determined through consensus among the authors before inclusion in our analysis. We excluded papers that did not handle the choice or decision context or focused on production methods or technical aspects of visual cues. We ended up with 146 papers illustrating the literature on visual design cues in a food decision context.

This set of papers was then read to give us a fair indication whether and how the effects of visual design cues have been tackled in food research and which gaps need to be closed.

To make sure that we did not miss out on relevant literature, we performed an additional structured literature search on each visual design cue (e.g., colour, shape, materiality) looking for papers combining specific keywords indicating the visual design cue character (e.g., colour) with the marketing aspect (e.g., product perception, choice) and the food aspect (e.g., food). To strengthen the discussion, we complemented these selected papers with papers that offer general theoretical insights from outside the food domain but can be applied to it.

The current paper thus offers a structured, narrative review of literature of the effects of visual design cues. For each visual design cue, we discuss the possible effects on behavioural outcomes and psychological processes. A table of contents can be found in Table 4. We illustrate the effects of each visual design cue with examples from previous research within various domains like visual perception, food economics, psychology, communication, behavioural sciences, graphic arts, architecture, engineering and agricultural economics. We highlight which future research (FR) possibilities exist in the context of food choices. An overview of the suggested research possibilities can be found in Table 5.

In the following paragraphs we discuss each visual design cue. We first discuss what it means and whether there are sub dimensions of this visual design cues. Then we provide examples of the manifestation of each visual design cue in the context of food choices and discuss previous research and future research directions for each visual each design cue. We discuss the effects of each visual design cue on “product perceptions and attitudes” and “other psychological processes” in separate headings.

## 4. Object Processed Cues

### 4.1. Colour

Colour is hue, lightness (also labelled: brightness or value [56]) and saturation of the perceived exterior layer of an object within the perceptual field [57]. Hue is the pigment. The universal colour spectrum contains the colours black, white, red, green, yellow, blue, brown, purple, pink, orange, and grey [58]. Lightness, brightness or value refers to its lightness or darkness [36]. A colour is bright in a scale from white to black. Saturation refers to the depth in a colour [59] or the intensity or amount of pigment in a colour [39].

In a food context, different colours can be applied to products, packaging, labels and logos. A multitude of research has investigated the effects of colour on behavioural outcomes and psychological processes. For reviews on the effects of surface colour we refer to [7] and [39]. Both reviews indicate that colours have metaphorical meanings that are often activated outside conscious awareness [60]. These metaphorical meanings can have an automatic influence [60] on biological responses, associations, emotional, cognitive, motivational reactions and behavioural outcomes [39,61]. Before we discuss several behavioural outcomes and psychological processes that the three dimensions (i.e., hue, lightness and saturation) of colours may induce, we discuss in brief the meanings and associations colours hold that could be relevant for a food choice context.

#### 4.1.1. Colour Associations

Specific colours hold referential meaning which is build up when individuals encounter pairings of colours with particularly meaningful messages, concepts, objects and experiences [39]. Through repeated exposures to colours, we have learned colour specific associations. For example, the colour red is a signal of warning, danger, prohibition and the need for vigilance [62]. Aslam [63] found that green and yellow are associated with luck and good taste. Green is also associated with nature [60,64,65], environmental conscious consumption and sustainability [66,67], ruggedness [39] and a healthy lifestyle [68]. Blue is associated with competence, purple with luxury and black with sophistication and glamour [69]. Black is also associated with the words ‘fatal’ and ‘poison’ [70] as well as with the concepts of dirt and toxicity [71]. Black and grey are associated with technology and sustainable products are associated with dull colours like green brown and white [72,73]. Greenleaf [74] noted that black and white can also be used to automatically evoke a sense of nostalgia. Black and white messages also convey timelessness and endurance, revealing the deeper meaning of the depicted content [75]. Meanings differ between cultures [76]. For example, red means unlucky in Chad, Nigeria and Germany, but lucky in China, Denmark and Argentina [63]. Interestingly, colours can mean different things in different contexts [50]. According to the colour-in-context theory, the effects and meanings of colours depend on the context in which colours are perceived [43]. Green can be associated with nature (in a real environment) or can signal money (in a marketing environment). Red signals either threat or opportunity depending on the context. It is associated with negative things like fire, blood, anger and danger [77] while it is also a biological signal of attractiveness [78] and ripeness of food [79]. Maier et al. [80] show that even infants have a preference for red in safe conditions and avoid red in threatening circumstances.

Colours can translate meaning and show quickly what a product is about or help identify which brand a product holds [81]. Marketers can shape consumer’s colour associations [39] like, for example, gender specifications (e.g., pink for girls and blue for boys). Consistently, using the same colour to communicate about intrinsic product attributes can also make this colour diagnostic and therefore allowing people to quickly identify important product attributes [82,83]. In some countries, the amount of fat milk containers is, for example, communicated by the colour of the milk bottle top (e.g., low fat version has a light blue bottle top; 83). By consistently using ‘light blue’ as a signal for ‘low-fat’ food people can clearly identify low-fat products. In addition, the learned association between the colour ‘light blue’ and the attribute ‘low fat’ is strengthened in people’s mind and this colour obtains meaning: light blue means/is associated with low fat [82]. These learned colour associations should be taken into account when one is planning to visually differentiate a brand by breaking with colour category norms [84]. People expect functional (sensory) products to have functional (sensory) colours like blue (red) [85] and breaking with the appropriate norm could be deemed ineffective for some product categories [84]. A good example is the failure of Crystal Pepsi: they applied clear colours instead of black to signal the absence of caffeine, [49] which could be attributed to unfulfilled taste expectations [39]. Colour norms for specific product categories and choice occasions as well as general hue preferences of people should be taken into account when making colour choices for food packaging, products, communication etc. [72]. Certainly, in the context of food choices, colour associations are crucial as they impact taste expectations, (un)healthiness perceptions, etc.

#### 4.1.2. Hue

##### The Effect of Hue on Product Perceptions and Attitudes

Package colours influence perceived **product quality** [35,64,86]. For example, packaging with contrasting colours are perceived to hold value-for-money products which are generally perceived to be of lower quality [87]. Products targeted to higher income classes, signalling higher price and elegance, are often packaged in cold, dark packages while light-coloured packaging is often offered to price sensitive people [88]. Green labels on beer bottles evoke higher quality ratings [89]. Also, product colour affects quality perceptions. In a beer context, dark beer is expected to be more expensive than pale beer [90]. It is unclear how quality perception (which is often assessed based on extrinsic product features [32]) is assessed when new colours (for example a brown tomato) are introduced on the food market and how this affects behavioural outcomes (FR 1)? Is quality perception lower than for ‘regular-coloured’ food products? How do people rate quality when intrinsic quality cues they normally use (like colour) cannot be used? Previous research suggests that when one’s knowledge is insufficient to make adequate evaluations of a product, different abstract elements can help decrease the ambiguity regarding what taste to expect that the product might have [27]. Are people relying on other intrinsic quality cues like smell, or are they looking for extrinsic quality cues like labels or other information provided by the company or store?

Package colours influence **taste perception** [35,91]. Package colour can be associated with a specific taste which can help rate a product when one cannot access it through experimentation [92]. The amount of orange colour on orange juice packaging increases sweetness expectations [91,93]. Food products in red packages were expected to be sweeter than those in green and blue packages [94] or white bowls [95]. Blue-to-green packaging was most strongly associated with a sour taste [93]. Green labels on beer bottles evoke higher taste ratings and indicate a dominance of fruity/citrus notes compared to a brown label [89]. Carvahlo and Spence [96] showed that taste expectations for coffee depend on the cup colour, the coffee type and congruent/incongruent pairings of colour-taste between cup colour and coffee type.

Product colour can set sensory and hedonic expectations regarding the likely taste and flavour properties of food and drink items (see for reviews [20,90,97,98,99]). Associations exist between red and pink with sweet, white and blue with salty, green and yellow with sour, and black and green with bitter [100]. In the context of food choice, red may still be connected to sweetness and green to sourness as many fruits become sweeter when they ripe and their colours transition from green to red [97]. Warm colours are associated with sweet flavour [91]. Red-coloured solutions were expected to be sweeter [101] or saltier [93] than green or uncoloured ones. Carvalho et al. [90] found that dark beer was expected to taste more bitter, taste stronger and have more body than pale beer.

Furthermore, it is unknown how taste of food products with unexpected colours (for example a purple cauliflower, green ketchup) is perceived and whether this affects behavioural outcomes (FR 2). Previous research shows that inappropriate colours can influence identification, and taste perceptions when the taste of the product does not fit the colour (for example grape taste with green colour) [97]. It is unclear whether coloured versions of common food products (e.g., purple cauliflower) influence taste expectations and hence whether experimenting with new food colours is an interesting marketing opportunity. How far can a colour of a food product be removed from the colour of the category representative? How do people react to a purple or pink cauliflower versus a light green or orange cauliflower?

Colours are used to assess **health perceptions.** Often warm colours are associated with unhealthy vice products like fast food [64]. Van Rompay and colleagues [102] showed that products in warm-coloured packaging were seen as less healthy than products in cold-coloured packaging. Similar results are reported by other researchers [103,104]. Tijssen et al. [104] found that people show strong associations between less vibrant, watered-down packaging colours and ‘healthiness’. People perceive a candy bar as more healthful when the calorie label is green rather than red or white [105]. Da Rosa et al. [93] found the opposite: packaging designed in a red-to-yellow (vs. blue-to-green) colour scheme showed higher averages of expected healthiness, despite the product category. Warning labels in black and grey were associated with less healthy products compared to warning labels in red and violet [68]. Further research could investigate whether these mixed results can be explained by the specific product categories the authors investigated or by the specific product colour (FR 3). De Temmerman et al. [106], for example, found that nutrition labeling containing colour indications of healthiness of a product especially affect behavioural outcomes for unhealthy food products rather than for healthy food products. Kapossa and Lick [107] found that the plate colour (black or white) on which pastries are presented influences greasiness perceptions of the food. Pastries in lighter colours like green, pink, lemon and off-white and light brown are expected to be greasier when presented on a black plate, while a dark brown pastry presented on a white plate is expected to be greasier. It seems like the contrast between the food colour and the plate colour affect food expectations: higher contrast results in higher greasiness. To date, it is also not clear whether food package or label colour induces inferences about for example caloric value or nutritional value. More reddish nuances in fruits and leaves generally indicates higher energy or protein content [108]. Foroni et al. [109] found that people attribute less energy to greener (vs. more red) food. Does the same apply for food packaging, label or logo colour?

Furthermore, it is unclear whether package, product, label or logo colour affects **sustainability** or **size perceptions** people hold and results in different behavioural outcomes (FR 4). Hagtvedt [110] noted that people view a darker (vs. lighter) packaged product being more durable, but less convenient, possibly because darker products are perceived as heavier.

Certain colours evoke more **positive attitudes** than others, but study results vary in which colours evoke the most positive reactions. Chattopadhyay et al. [76] show a preference for the hue of blue across different cultures if hue is made salient. Kareklas et al. [111] found a general preference for light over dark colours. Greenleaf [74] suggests higher liking of monochrome colours (black and white, BW). Da Rosa et al. [93] show that red to yellow and blue to green package colour schemes were preferred over greyscale packaging when the product category was not identified. Also, Rebollar et al. [92] found that people prefer warm-coloured packaging over other coloured packaging. Mixed results can be explained for example by the time of day meaning that monochrome images could be preferred during night-time, while non-monochrome colours are preferred during daytime [74]. Also colour preference could be steered by fashion trends. Colour preferences also depend on type of products: sport cars may benefit using the colour red, while this could harm evaluations of high-priced durables [112]. Further research should look into the factors that explain when and why specific hues are liked and whether this translates in behavioural outcomes (FR 5).

Interestingly, colours affect product liking especially when one lacks the motivation to process [113] suggesting that colours can especially be used to communicate about products that people are less interested in [50]. Peng and Jemott [114] found that using more arousing colours (e.g., red and orange) and using a variety of colours enhanced a food image’s aesthetic appeal and likeability. On the contrary, Gorn et al. [115] found that hue had no effect on liking in an advertising context. Carvahlo et al. [90] show that people expect to like pale beer more than dark beer prior to tasting. Schifferstein et al. [110] found that the colour of the background on which fresh food products are presented (for example plates) influences the attractiveness of products. When contrast between the background colour and the food product colour is high, the attractiveness of the fresh product is enhanced (except for carrots). Bix et al. [116] found the opposite: products and backgrounds in same or analogous colours attract more attention and are liked more than food products and background in contrasting colours.

##### The Effect of Hue on Other Psychological Processes

Colours are often used to attract **attention.** Colours guide attention towards motivationally important objects [117]. Warm colours attract attention and especially attract impulsive buyers [118]. For example, red attracts more attention than green [119] especially in an emotional context [120] which can provide competitive advantage [121]. For example, red packaged products like coca cola are more visible and could, therefore, be sold in greater quantities. The colour red is also used in colour coded nutritional labels. Previous research shows that using a red colour affects purchase behaviour in the sense that people are inclined to choose less unhealthy products labelled with a red colour compared to when this red colour is not present (e.g., only caloric indication) [106]. Further research is necessary on the effects of colour coding in nutritional labels on behavioural outcomes and how they can be explained (FR 6). For example, in which contexts and for which product categories does colour coding stimulate healthy or decrease unhealthy consumption? Are colour codes deferring attention from more detailed ingredient information making people overly—but wrongly—confident in their ‘healthy’ choices? Does colour coding evoke other cognitive, affective or motivational reactions steering behavioural outcomes?

Mixed findings exist on which colour attracts most attention. Young [122] found that red warning labels were more noticeable than black labels. On the contrary, Cabrera et al. [68] found that black warning signs were easier to find than red warning signs. Also, Bialkova and Van Trijp [123] found that monochromatic directive Front-of-Package (FOP) nutrition labeling schemes capture more attention than polychromatic logos. Cabrera et al. [68] suggest that these contradictory results can be explained by the context in which the labels are shown. Important to note is that visual salience in general depends on the context: a feature is visually salient when it stands out compared to their surroundings [49]. So, the attention-grabbing character of a colour cannot be measured in isolation. A black and white package will attract attention on a shelf filled with coloured packages, even though the latter are in general more visually salient and promote attention and memory. Text on a package will only be salient when the text colour is considerably different from the background [124]. Similarly, some contrast will be more visually salient than others. Frey et al. [125] found—in a natural scene context—that contrasting colours red and green results in higher salience (and hence attracts more attention) than the blue–yellow colour contrast. Future research could focus on which specific colours and colour combinations attract attention in food choice contexts and whether this impacts behavioural outcomes (FR 7). For example, will a black and white display in a fresh produce assortment holding especially food products with warm colours attract more attention than a colourful display using either warm, cold or a combination of warm/cold colours [126]? Does more attention imply more positive food perceptions, attitudes and behavioural outcomes?

Colours can facilitate specific **cognitive reactions**. Blue for example, can stimulate creativity [127] while the colour red facilitates analytical skills [118]. Colours affect cognitive performance [49]. Greenleaf [74] suggests that monochrome images (for example black and white images) could be more easy or more difficult to process than colour images depending on the familiarity with the objects displayed in the image. In this context, it could be interesting to investigate the possibility that specific colours direct attention to different aspects of products (FR 8). Lee et al. [128] found that black and white stimulate individuals to focus on the abstract, essential, and defining features of a stimulus (high-level construal), while colour makes individuals focus on the concrete, idiosyncratic, and superficial features of a stimulus (low level construal). They show that a black-and-white (vs. coloured) visual on packaging enhances the perceived importance of primary (vs. secondary) product features leading people to prefer products with superior primary relative to secondary features [128]. Although colour promotes attention and memory, colour can also distract individuals from attending to more essential and primary product features leading to a greater willingness to pay premiums for products with unnecessary and superfluous features [128]. They also suggest that black and white may allow people to transcend the particulars of the moment and focus on bigger and broader generalities indicating for example that they would focus more on the why of consumption rather than on the specific how of consumption. Future research could investigate whether specific hues focus attention to specific product features of food (e.g., healthiness, fat level, calorie content, energy level) due to an activation of a low or high construal level and how this affects product perceptions, attitudes and behavioural outcomes. For example, colours that trigger high construal could stimulate healthy or sustainable food choices (which satisfy long-term goals).

Furthermore, cognitive processes other than construal level like creativity could be stimulated by specific colours (FR 9). Specific colours could affect processing difficulty or divergent/convergent thinking. Dark colours are more concentrated so they could “close up space” triggering convergent thinking, while light colours—that could be seen as more open—could evoke divergent thinking. Furthermore, specific colours could evoke analytical skills which can help people to better differentiate between for example high-quality and low-quality foods or foods differing in sustainability or healthy level.

Specific colours evoke specific **affective reactions** like feelings (i.e., happiness, anger, fear) and mood states (i.e., relaxed vs. excited) which in turn affect for example how someone evaluates a store atmosphere, an in-store ad or a food product, whether one is willing to buy and the price one is willing to pay (for a review see [39]). For example, longer-wavelength hues (e.g., warm colours like red [129]) induce excitement [64,102,115,130], arousal [60,109,112,115] and warmth [60]. Shorter-wavelength hues (e.g, cold colours like blue, green, purple) are calming and refreshing and evoke feelings of peacefulness and relaxation [32,112,115].

Although research identifies the link between specific hues and specific feelings and mood states, less is known about what the downstream effects are of these feelings and mood states (FR10). Bellizzi and Hite [112] did found that the background colour blue resulted in more purchases, less purchase delays, and a stronger tendency to go shopping, possibly due to the emotional effect of the colour blue. Future research could investigate how specific feelings and mood states evoked by specific hues affect product perceptions, attitudes and behavioural outcomes. Does food taste differently when you feel aroused by logo, package or food colours? Is perceived food quality higher or lower when specific logo, package or food e colours evoke specific feelings or mood states? Do people choose more healthily when they experience a sense of vitality (induced by logo, package or food colours)? Do all these effects depend on the type of product (for example, healthy vs. non-healthy, fresh or processed)?

In the context of food marketing, the effects of black and white versus colours is an interesting research area. Food packaging is often colourful, but some manufacturers use black and white packaging. In a related way, often private labels often use less-extended colour schemes than manufacturer brands. It is unclear whether black and white images trigger more or different emotions than coloured images. Some authors argue that colour evokes more emotions than black and white [39,60,64] while others find no differences in arousal [130] or emotional experience [128]. It is possible that contextual differences can explain these mixed results. Future research could investigate whether black and white trigger more or different feelings and mood states than colour when making food choices (FR11). Do these emotional reactions depend on the context? What are the downstream effects (in terms of food perception, attitudes and behaviour) of black and white versus colours in a food context? For example, do black and white package, logo or image colours trigger other feelings and mood states and therefore more unhealthy or healthy choices?

Moreover, colours can evoke specific kinds of **motivational reaction**. Colour can serve as an automatically and rapidly processed affective prime and facilitate approach—or avoidance-oriented psychological processes [43]. Red elicits a general avoidance motivation for example [61,131] which translates in a shopping context to less purchases (compared to the colour blue [112]). Red also makes people highly engaged to win [132]. This could be particularly interesting in a context of food products that most people try to avoid like insect-based products and ugly looking or wonky fruit and vegetables. Future research could investigate whether a specific package, food or logo colour could evoke an approach motivation making people more willing to buy these ‘disgusting’ evoking food products (FR 12) [133]. Can specific package hues evoke approach motivations overcome disgust for specific types of sustainable food like insect-based food, cultured meat, ugly looking food and vegetables and trigger positive behavioural outcomes? Does this effect depend on the level to which food appearance differs from the norm?

#### 4.1.3. Lightness

##### The Effect of Lightness on Product Perceptions and Attitudes

Limited research exists on the effect of lightness of colour on product perceptions. For example, light and artificial colours generate low **quality** judgments [134]. Food is seen as more **healthy** when colours are lighter [92]. According to our knowledge, no research to date exists on the effect of colour lightness (of specific food product, food packaging, label and logo lighting) on taste perceptions, sustainability and size estimations (FR 13). Does this affect behavioural outcomes? For example, do specific colour lightness levels make people rate products as more tasty, more sustainable or larger and are therefore more likely to make them choose these products?

Gorn et al. [115] show more positive **attitudes** for colours with higher (vs. lower) levels of lightness. Hemphill [135] shows that lighter colours evoke more positive reactions compared to darker colours.

##### The Effect of Lightness on Other Psychological Processes

In general, research on the effect of lightness of colours on psychological processes besides product perceptions and attitudes is scarce. Some research suggests that lighter [136] colours (in comparison to its surroundings) are more visually salient which could increase the chance that a product is noticed and considered. Future research could investigate further how lightness of product, packaging, label and logo colour affect attention, affective, cognitive and motivational reactions and which downstream effects they have on product perceptions, attitudes and behavioural outcomes (FR 14). For example, does colour lighting affects construal level, self-control etc. which in turn affects food choice?

#### 4.1.4. Saturation

##### The Effect of Saturation on Product Perceptions and Attitudes

To date, limited research investigates the effect of product colour saturation on product perceptions and attitudes. Tijssen et al. [104] found that a combination of hue and saturation (for example high-saturated red) increased **taste perception** (i.e., perceived flavour intensity and sweetness and creaminess in a dairy drink). The use of colour saturation in assessing taste and **quality** could be an interesting future research area (FR 15). What is the impact of product colour saturation on quality perception? Does this affect behavioural outcomes? Is the effect of colour saturation on quality and taste perception different for different products (for example fresh produce, fruit juices, dairy drinks?) and for different ripeness levels of fresh produce? A deep red tomato could be assessed as higher quality or better taste compared to a pale red tomato. However, this could depend on the ripeness of the fruit or vegetable. The difference between a light green and darker green (unripe) tomato probably does not affect quality or taste assessment since one knows the colour is going to change, while for a ripe (red) tomato, the saturation does matter as a quality or taste cue? Does saturation affect **healthy**, **sustainable** or **size** perceptions?

Saturation of the package colour could influence product perceptions. Becker et al. [22] found no differences in taste perceptions of yoghurt packaged in low- and high-saturated green packages. Increasing colour saturation of package container increases size perceptions of that container and increase amount of the product that is chosen to fill this container [56].

Meadt and Richerson [23] found for example that people perceive products in vivid highly colour saturated food packaging as less healthful than the same product in muted, less colour-saturated packaging. This could possibly be explained by an artificial association people hold with vivid colours. Future research could look further into the effects of package colour saturation on food perceptions and resulting behavioural outcomes (FR 16). For example, do highly saturated packages signal high or low quality, taste, healthy or sustainable character? Does this depend on the food category? Does it lead to increased purchase intentions, food choice or willingness to pay?

Gorn et al. [115] show higher **liking** for colours with higher (vs. lower) saturation levels. Hagtvedt and Brasel [56] found similar to [137] that higher saturation indeed makes attitudes more favourable—and willingness to pay is higher—but only for products with high (low) saturation when usage goals call for large (small) size.

##### The Effect of Saturation on other Psychological Outcomes

Again, limited research investigates other psychological processes than product perceptions of colour saturation. Higher saturated colours capture **attention** [18,56] and could increase memory since they increase **arousal** [115]. No research investigates the effect of colour saturation on **cognitive and motivational reactions** and resulting food perceptions, attitudes and behavioural outcomes (FR 17). It is possible that lower-saturated colours give rise to convergent thinking making people less open to try out new foods. Possibly self-control is lower in case of higher-saturated colours making people more prone to eat unhealthy. Colours with higher saturation levels induce **feelings** of excitement [115]. Hagtvedt and Brasel [56] argue that higher-saturated colour stimulate arousal. Highly saturated images also evoke stronger emotions [137]. These strong positive emotions could possibly result in more positive attitudes by for example a halo effect [138] or the what is beautiful is good heuristic [139].

### 4.2. Shape

A second, important visual design cue is ‘shape’ which refers to the geometrical property of the cue. Sample et al. [7] (p.409) define shape as: *“The perceived space occupied by an object in the perceptual field as compromised by the outer boundaries of that object.”* It can range in dimensionality (i.e., height, width, length), the unity of the shape (i.e., an object’s perceived cohesiveness as allowed by segmentation and occlusions), demarcation lines (i.e., the outer boundary that contains the entirety of a perceived object) and shape contrasts (i.e., the deviation of a perceived object from context or consumer experience) [7]. Adaval et al. [50] talks about shape in terms of the “ratio” between the lengths of the sides. In addition, Greenleaf and Raghubir [31] refer to its range in complexity, curvature, congruence, and completeness. Complexity refers to dimensionality, form, regularity and clutter; curvature refers to the angularity, circularity and convergence of the shape; congruence refers to symmetry, planned distortion, stability and centrality to and completeness refers to synthesis, amount of information and incomplete pattern and shapes. For more information and examples, we refer to [31]. Some commonalties exist between the typologies of Sample et al. [7] and Greenleaf and Raghubir [31]. Curvature is similar to what is called ‘demarcation lines’ by Sample et al. [7]; both refer to dimensionality.

In a food context, relevant geometrical properties contain the ‘dimensionality’ of the shape of the food product, packaging or label (e.g., wide or long package, big or small label), the ‘demarcation’ (e.g., square, rectangle, circle, curves, etc.) of the food product, packaging or label, the shape ‘contrast’ (e.g., is the shape different than expected or than other food products/packaging/labels in the context?), the ‘completeness’ of the food (e.g., is the shape of food product a full unit? or cut in half?), the shape complexity and shape symmetry. Complexity and symmetry are also related to aesthetic characteristics of the visual design cue and are discussed under aesthetic appeals (Section 4.3.1 and Section 4.3.2). Shape contrast is not discussed as a geometrical property as such but is discussed as a part of other geometrical properties. Shape contrast is seen as a shape that deviates from the context or from experience. Hence, this could entail a package size that is bigger or smaller than other packages, a package demarcation that is different to what one is used to, a product that is less complete than one is used to etc.

Previous research on the effects of shape characteristics of products, packaging, labels or logos especially concentrated on the effect of dimensionality and demarcation on product perceptions, attitudes and behavioural outcomes. Limited research concentrated on the effects of food shape completeness as visual design cue and the effects of dimensionality, demarcation and completeness on other psychological processes.

#### 4.2.1. Dimensionality

##### The Effect of Dimensionality of Shape on Product Perceptions and Attitudes

Similar to colours, shapes also convey meaning [140]. Past experiences with shapes affect product expectations through associative learning [91]. The shape of the packaging influences people’s expectations concerning the product’s functional, experiential and sensory (textural) attributes [91]. For example, the dimensional shape of a container in which food is presented influences specific taste sensations [141]. People can use dimensional aspects of the physical shape of a food product or container to estimate **size [142,143,144,145]** which in turn can affect preferences [145,146,147]. Krider et al. [145] discuss a psychophysical model how people use heuristic processing to judge areas by making effort–accuracy trade-offs leading to systematic shape and size-related biases which influence how much a consumer is willing to pay. Raghubir and Krishna [146] show that people overestimate product sizes because they anchor their perceptions on the most elongated or primary dimension and do not sufficiently adjust for a smaller secondary one [145]. People do not take devote time or energy in calculating actual product size and rely on heuristics that help that make decisions based on one or two salient dimensions without adjusting for the remaining dimensions [142]. The most visually salient dimensions drive evaluations [148]. For example, Krishna [149] showed that the height dimension tends to be most salient evoking an elongation bias (i.e., tall containers seem larger than shorter/wider containers. Raghubir and Krishna [150] and Wansink and Van Ittersum [147] found that elongated packages are rated as containing more volume (and hence food), which could create the illusion that one consumes less from a more elongated container. Szocs and Biswas [150] confirmed previous research stating that longer, thinner-shaped products were rated larger than shorter, thicker-shaped products. These longer products were also rated to contain more calories; no differences in **taste** perceptions were found. Further research could investigate whether shape dimensionality does affect taste perceptions and behavioural outcomes in particular contexts. Furthermore, it is unclear whether shape dimensionality affects **quality** expectation or assessments, **healthiness** or **sustainable** perceptions of the food product or package. For example, bigger containers could signal lower quality, healthiness or sustainability through an association of “less is more”. Koo and Suk [151] show that longer, vertical packages (vs. wider, horizontal packages) are perceived as having fewer calories and are associated more with healthiness compared to a horizontal package. This vertical or horizontal package shape could create a health halo causing people to think that the food is healthier than it actually is [37]. We propose (FR 18).

Research suggests that a package’s shape is a critical way for a brand to differentiate itself and sales and profits can be influenced by small changes in package shapes [152]. Objects with specific geometrical properties induce more positive **attitudes** [31]. Raghubir and Greenleaf [153] show that people have a particular preference for ranges of ratios (i.e., the lengths and sides of a rectangle) that can be aesthetically pleasing and these ranges depend on the context (e.g., more serious—detergent/soaps—vs. less serious—cookies, cereals—product categories). Product and packaging shapes have been found to influence preferences so that people prefer more down-sized products [143].

##### The Effect of Dimensionality of Shape on Other Psychological Processes

Limited research investigated the effect of dimensionality on other psychological processes besides product perceptions and attitudes. Bigger labels attract more **attention** [68,154] which can increase the chance that they are being bought. The results of Hummel et al. [141] suggest that people have more thoughts about wine when served in a bulbous glass suggesting that the dimensional package shape can affect **cognitive reactions**. Future research (FR 19) could look further into the effects of package size on attention, cognitive, **affective and motivational** reactions and how they can explain downstream effects on food perceptions, attitudes and behavioural outcomes. For example, do smaller food packages induce convergent thinking? If so, smaller food packages could be evaluated on different food cues compared to larger packages. Do bigger sized packages evoke higher arousal or specific emotions like fear? Bigger sized packaging could also induce approach behaviour and decrease self-control because of our inherent motive to gain and maintain respect or prestige [44] resulting in more positive behavioural outcomes.

#### 4.2.2. Demarcation

##### The Effect of Demarcation of Shape on Product Perceptions and Attitudes

Shapes metaphorically communicate their key characteristics and affect product perceptions [155]. We have learned to associate meanings to particular shape demarcations (e.g., angular shape, circular shape, hourglass shape, etc.) and specific product perceptions. For example, borders with specific shapes enclose warnings which influence risk perception [156]. Hence, these shapes become associated with risk. Riley et al. [157] for example found that pointed shapes, especially a triangle on its vertex, tend to be more associated with danger than rounded shapes.

Demarcation influences **size** perceptions: triangles have been generally found to be perceived to be larger than circles and squares while mixed results exist on perceived size differences between squares and circles (see [145] for an overview). This could also lead people to expect products to be of higher or lower **quality** when presented in an angular shaped package versus a rounded package. This bias is partly driven by disproportional attention to salient but misleading physical properties of the stimulus [142,145]. Folkes and Matta [142] also found that bottles shaped with unusual forms or contours (i.e., shape contrast) are rated larger. They argue that shapes that attract more attention are perceived as having larger volume than same-sized containers that attract less attention and are seen as a better buy [142]. Folkes and Matta [142] also suggested that this increased attention effect disappears when individuals get used to a product. In this context it could be interesting to see whether unusual products or unique shaped product packaging can be associated with a rarity and, therefore, a good buy (FR 20) [50].

Demarcation influences product assessments and sensory qualities [22,158,159]. The specific demarcation also influences **taste** complexity ratings [160] and the intensity of taste sensations [161,162]. For example, Becker and colleagues [22] found that angular packages are associated with a more intense taste than rounded packaging and are expected to be more expensive. De Bondt et al. [163] found that products in human-like shaped packages (V-shape and hourglass shape) were rated higher on perceived tastiness than products in angular, rectangular or oval packages. Food shape can also potentially influence oral haptic aspects (i.e., how food is perceived in the mouth [150]).

Specific shapes are often matched with specific tastes (for reviews we refer to [27,158,164,165,166,167]). Ngo et al. [157] found for example that angular shapes are generally associated with bitter and sour or “sharp” tastes, while round shapes are more associated with sweet and rich tastes. Also other researchers confirmed that rounded packaging are associated with sweet [167] or fruity [162], smooth (creamy) taste while angular shapes are associated with sour tastes [28,167,168,169]. Manipulating window shape (rounded vs. angular) did not have an effect either on sweet or sour expectations [170]. By contrast, Da Rosa et al. [93] found that angular packaging evoked higher associations with sweet tastes than did the rounded packaging. They explain their different results due to the lack of definition of the product category in their study. In that case, people have no sufficient knowledge to evaluate the taste of the product and have to use different abstract elements to make their decision than the attributes they would use when they had identified a specific product category. The association between product or package roundness and sweetness of a product could explain why round packages are liked more [171]. Curved shapes are also associated with vanilla aroma and angular shapes with citrus [172]. Carvalho and Spence [129] found that a tulip-formed coffee cup is associated with stronger aromas, while sweetness and acidity of coffee aroma is higher in a split cup. Similarly, Van Doorn et al. [173] narrower coffee mugs are associated with more aroma and shorter mugs were associated with more bitter and intense coffee, while wider mugs were expected to hold sweeter coffee. It is not known what the effect of unusual packaging shapes is on taste. Future research (FR 21) can for example investigate whether healthy or sustainable food products can be packaged in unusual or unique shaped containers to enhance taste perceptions and result in more positive behavioural outcomes. Earlier research seems to suggest negative effects: individuals have expectations about packaging shape so that packaging differing from the prototypical product attract more attention but are evaluated more negative [142,174]. This would suggest that people would not be inclined to have positive taste perceptions (and hence buy) food in atypically shaped packaging.

Related to the link between round packaging and sweetness perceptions, Fenko et al. [101] found that (muesli) cookies are perceived as being more **healthy** in angular (vs. round) packaging. By contrast, Da Rosa [93] found that cereal and buttery cookies are perceived as more healthy in a round versus angular packaging. Warning labels in octagonal shapes were associated with a less-healthy product compared to warning labels with a triangular shape [68]. Furthermore, Yarar et al., [175] found that packaging resembling humanoid shapes increase healthy ratings of food especially for women who have higher body mass index (BMI).

Research on “misshapen” produce suggests that people tend to dislike products that do not resemble the norm. People are less willing to sell, buy or consume suboptimal (on the basis of appearance standards in terms of weight, shape or size) or oddly shaped foods with aesthetic or cosmetic imperfections [176,177,178,179,180,181]. People are willing to buy food suboptimal in appearance when sprayed with pesticides [182], when the deviation from the norm was moderate (vs. extreme) [183], when authenticity/sustainability positioning is used [184] or when a high discount was given [179,185]. This willingness and the optimal way to increase it also depends on the type of misshapenness [179]. For example, willingness to buy an apple with a spot is lower than for an oddly shaped cucumber [179]. Cooremans and Geuens [186] explain the preference for ‘normal’ shaped products by categorization theory (‘normal is good/abnormal is bad’, [187]) and the “what is beautiful is good” [188] or “the ugly is bad” [189] stereotype. Cooremans and Geuens [186] found increased risk perceptions and decreased taste, health and convenience perceptions and purchase intentions for misshapen fruit and vegetables. Also Loebnitz and Grunert [190] found that risk perceptions are higher when the shape of the product deviates more from the norm. Powell et al. [133] found that disgust propensity decreases taste and naturalness perceptions and visual appeal of atypical-shaped fruit which in turn affected a lower willingness to pay. This is in line with other studies showing that disgust is an important determinant of food choices [191,192]. Powell et al. [133] suggest that interventions enhancing taste, naturalness and visual appeal of these atypically shaped fruit and vegetables to increase willingness to consume. In this respect, Cooremans and Geuens [186] found that anthropomorphizing vegetables (i.e., the tendency to imbue non-human objects with human-like characteristics, intentions, and behaviours [193], p. 864) like adding for example eyes, mouth, and arms) resulted in an increased choice of misshapen vegetables while this increase was not found for ‘normal’ shaped vegetables. They argue that by shaping abnormalities as body parts or adding a smiling face may render the abnormal shape cute (rather than weird), thereby overriding an “abnormal is bad” heuristic. Also, the presence of positive human features (e.g., a smile) can make the product more attractive, preventing a negative carry-over effects from an “ugly is bad” heuristic. They suggest that these positive features can increase product attitudes and purchase intentions by activating a positive human schema [164]. Aggarwal and McGill [194] and Darriet [195] also showed that attitudes can increase by using anthropomorphic factors (in ads) of ugly food by creating a connection between consumers and products helping consumers feel greater moral care and trust toward the product [196]. Also Shao et al. [197] confirmed that an ad showing anthropomorphic elements could increase purchase intentions for ugly-looking potatoes. It is possible that this effect would only hold for products with predictable/symmetric appearance and not for those that have a less symmetric appearance [186], for items that you peel (e.g., tangerine) and consume as is (e.g., strawberry). Lombart et al. [198] already showed in a virtual store that quality perceptions decrease when misshapenness increases. Further research is necessary to investigate when and why unusual food shapes affect quality, taste, health and **sustainability** perceptions and how negative effects can be countered (FR 22).

In general, people prefer rounded shapes (e.g., circles) to more angular shapes (e.g., triangles or stars) [172,199,200,201]. Wang and Zhang [202] found that a downward triangle is perceived as negative and a circle is perceived as positive. However, they did not find an influence on behaviour possibly because these shapes are rather abstract threats rather than real threats. Westerman et al. [203] manipulated the shapes of labels appearing on bottles of water and vodka and found that rounded labels are preferred. People also display higher intentions to buy packaging displaying rounded shapes (see [204] for a review). Round packaging is also preferred over squared or angular packaging [28,93,205]. However, Fenko et al. [103] found that cookies in an angular package were expected to taste better than cookies in a round package and hence purchase intentions were higher for cookies in angular packaging. Also [37] found a preference for straight over curvy packaging, probably because people were more familiar with the straight packaging. Yang and Raghubir [206] shows that container shape affects the number of units of a product that a consumer purchases. Krider et al. [145] showed that people buy a higher amount of round-shaped versus rectangular-shaped packages. da Rosa et al. [93] indicate the importance of the product category in this context. In one of their studies, in which the product category of the stimuli was not defined, they did not find a preference for round shapes. When the product category was not defined, people do not know the shapes that are typical for this product category (due to higher exposure) and cannot bring to mind experiences and sensations felt when consuming those brands [207] which then defines the shape that they prefer [37]. It could be interesting in this context to investigate whether and why specific package demarcations can be used as an optimal intervention to increase healthy or sustainable food consumption (FR 23). Van Kleef et al. [208] already showed in a children’s sample that shaping whole wheat bread as funny increases choice while no effects of fun shapes were found for (more unhealthy) white bread.

##### The Effect of Demarcation of Shape on Other Psychological Processes

Special product or packaging shapes deviating from expectations or other packaging shapes (i.e., shape contrast) attract attention [54,141]. Folkes and Matta [141] showed that containers which attract attention, evoke more thoughts and are rated to contain more volume especially when people like the product within the container. Certain shapes also attract more **attention** due to their threatening character. Lee and Shumann show that atypical-shaped products that are moderately inconsistent with the norm can increase cognitive processing. As a result, individuals who want to resolve the incongruity may activate positive thoughts [209]. Further research is necessary to investigate whether and how atypical shapes elicit specific **cognitive reactions** and whether they can explain behavioural outcomes (FR 24). Do people focus on concrete product attributes (rather than abstract) when they are confronted with atypical shapes? Does it induce creativity? Do atypical shapes elicit a higher amount of thoughts? Also, no research has been done on the effects of demarcation on **affective and motivational** reactions and resulting behavioural outcomes. Do atypical shaped containers or food products elicit specific feelings of arousal, disgust, or avoidance motivations and hence lower behavioural outcomes? Do rounded or angular packaging induce self-protection or status seeking?

#### 4.2.3. Completeness

##### The Effect of Completeness of Shape on Product Perceptions and Attitudes

Some research has investigated the effect of incomplete products on product perceptions, attitudes and behaviour. Incomplete product means that the shape of the product does not appear to be a full product [210] like, for example, a sandwich that is already cut in half. Sevilla and Kahn [10] found that incompletely shaped products are rated to be **smaller** which could lead to increased consumption quantities of this incompletely shaped product. They explain this by the numerosity heuristic which suggests that people rely on the number of units into which a stimulus is divided and tend to underestimate other important aspects, such as the size of the units [211]. People perceive incompletely shaped products to contain less quantity than their equally sized, completely shaped counterparts [210]. Sevilla and Kahn [210] argue that the effect of completeness on quantity perception is strong because it is related to a motivational need for completeness [212]. The effect only disappears when the reason for incompleteness is made salient (e.g., Swiss cheese, bagel). That is, for some product categories incompleteness is inherent to the product (e.g., a bagel contains a hole) so the incomplete character of the product would have less effect on product perceptions, attitudes and behaviour. Research could further investigate whether incomplete products influence **quality, taste, health or sustainability** perceptions and **attitudes** and what the downstream behavioural outcomes are (FR 25).

##### The Effect of Completeness of Shape on Other Psychological Processes

No research to date has investigated the effects of shape completeness on **attention, cognitive, affective and motivational** reactions and how can they explain behavioural outcomes (FR 26). It is possible that incomplete products attract more attention and evoke more thoughts. The incomplete character of the product or package could also stimulate divergent thinking or concrete processing since it attracts attention to the attribute level of the product or package. People could be motivated to avoid incomplete food products or reduce risk seeking since incompleteness signals imperfection.

### 4.3. Aesthetic Cues

Adaval et al. [50] does not provide any specific definition of ‘aesthetic cues’, but he refers to the following aspects as examples of aesthetic cues: typeface, the use of art (art-infusion), balance, harmony, proportion (i.e., the golden ratio and the rule of thirds). The term aesthetics is based on the Greek word ‘*aesthesis*’ which means: the perception from the senses, feeling, hearing and seeing. Later on, aesthetics was defined as ‘*perfection of sensate cognition*’ [213].

Aesthetic design plays an increasingly larger role today in creating consumer desire for products [214]. Firms are shifting their differentiation efforts away from concrete product characteristics towards less tangible features such as aesthetics. The question, however, of what is or is not an aesthetic appeal seems hard to answer. Deng et al. [215] investigate people’s preferences for colour combinations as a manifestation of what is aesthetic. Veryzer and Hutchinson [216] identify unity and prototypicality as important visual aspects of design that trigger aesthetic responses. The typology of Phillips and McQuarrie [52] describe visual structure (e.g., juxtapositioning, fusion and replacement) as an element of visual aesthetics since it defines the level of visual complexity. Aesthetic elements relevant at the point of purchase are (1) balance and complexity, (2) symmetry (3) unity, and (4) prototypicality. Unity is discussed under the heading “shape completeness” (Section 4.2.3) while prototypicality is related to shape contrast or the fact that a shape is different from what one expects or is used to (see Section 4.2). We also discuss research on aesthetic elements that are used in point of purchase offline and online (social) media: cartoons, drawings, photographs and food porn.

#### 4.3.1. Balance and Complexity

Central to arts are balanced and complex design. A visual representation is said to be balanced when its components and their qualities are organized around a balancing centre so that they appear to be anchored and stable. There is considerable empirical evidence that balance influences the immediate and sustained perception of an artwork as well as its aesthetic qualities [217]. In visual arts, complexity increases with a growth in the number of elements within a composition. Researchers [87,217,218] reasoned that the effects of complexity and balance will hold for food plate presentations too (e.g., self-serving salad bar, ready-to-eat food boxes, food pictures on food-delivery websites).

##### The Effect of Balance and Complexity on Product Perceptions and Attitudes

A moderate degree of complexity increases liking for visual stimuli [219,220]. Zellner et al. [217] show that balance and complexity (i.e., colour vs. monochrome) interact in their effect on **attractiveness** of the food presentation. Colour enhanced the attractiveness of the balanced presentation but not of the unbalanced presentation. The enhancement of attractiveness of the balanced presentation by the addition of colour is probably due to the colour adding complexity as predicted by studies of artwork [221]. Zellner et al. [217] also found that colour and balance influenced the choice to try the food. Participants were more willing to try food presented in a balanced arrangement or from a monochrome presentation. These results are not in line with the attractiveness ratings where the colourful balanced presentation was rated most attractive. These results suggest that although people might find food on a plate to be aesthetically pleasing, they might not choose to try that food. The fact that subjects were more choosing to try the food in a monochrome-balanced presentation might be the result of neophobia (i.e., the extent to which individuals are reluctant to try novel foods, food products, dishes, cuisines [222]). People are much less willing to try foods which are unfamiliar [223]. Further research could look into which combination of balance and complexity of food presentation and packaging increases food attitudes and behaviour and why this occurs (FR 27). Orth et al. [224] investigate the effects of elaboration (i.e., complexity) and harmony (i.e., balanced design) in the context of product packaging. The authors show that harmonious packages are liked more and are expected to be higher in quality and, therefore, to be higher in price too. Orth et al. [224] also show that more elaborate (i.e., more complex) package designs evoke higher **quality perceptions** and price expectations. It is unclear whether balanced and more complex packages affect other product perceptions like food **taste, healthiness, sustainability or size**. Zellner et al. [217] call for further research on the effects of artistic design elements such as balance and complexity in different contexts of food choices.

##### The Effect of Balance and Complexity on Other Psychological Processes

According to Orth et al. [224] harmonious or balanced packages are liked more due to **processing fluency**. People must make less effort to process harmonious stimuli than contrasting stimuli, and this reduction in effort leads to more favourable outcomes. Further research is required on which effects will hold in a food choice context and whether these effects are moderated by food type (i.e., healthy vs. unhealthy food) (FR 28). Orth et al. [224] also suggest that more complex designs evoke abstract comparisons while less complex designs could evoke a concrete mindset. They suggest that more complex designs evoke higher arousal and engagement leading towards more positive attitudes [224]. Other questions remain. For example, do complex and balanced food presentations and packaging attract more attention? Do they evoke status seeking or protection motivations? Why do we like complexity? Does it motivate us to process information?

#### 4.3.2. Symmetry

Symmetry is a complex visual configuration, deriving from mathematics and geometry [214]. It can be defined as a transformation that preserves equal geometric distance in space [225]. An example is mirror symmetry where one half is the mirror image of the other half. These halves are said to be separated by a symmetry axis. We found limited research about this visual design cue in the context of food choices.

##### The Effect of Symmetry on Product Perceptions and Attitudes

Bigoin and Lacoste [214] show that symmetrical design of information items around the vertical axis on food product packaging (i.e., orange juice, chocolate bars, biscuits and pasta) positively impacts consumption experience and purchase intentions. Previous research shows that the positive effects of symmetry depend on product value (e.g., functionality, quality and ease of use) and visual complexity [226]. No research to date investigates the impact of symmetrical designs of food packaging and food products on **product perceptions** and resulting behavioural outcomes (FR 29). Do asymmetrical designs bias size estimations? Do they decrease taste and quality expectations because they differ from the norm? Do asymmetrical products seem less healthy and sustainable? Would people, for example, prefer a perfectly symmetrical strawberry above less symmetrical strawberry or would this seem unnatural? Would processes such as perceived naturalness, perceived authenticity or credibility impact the effectiveness of symmetry? Does our preference for symmetry depend on food type (healthy vs. unhealthy; processed vs. unprocessed foods)?

##### The Effect of Symmetry on Other Psychological Processes

The positive impact of symmetrical disposition of information around the vertical axis on food product packaging (vs. non symmetrical disposition of information) is driven by a decrease in visual complexity which positively impacts **processing fluency** [214]. Again, it could be interesting to find out whether other **attentional, cognitive, affective and motivational reactions** are triggered by symmetrical packaging or products and whether this can explain food perceptions, attitudes and behavioural outcomes (FR 30). Does symmetry always lead to higher processing fluency? Or does this depend on the type of media: packages, plates, or products themselves? Do symmetrical food packages or products increase self-control? Do asymmetrical food packages increase attention and arousal and hence result in more positive behavioural outcomes?

#### 4.3.3. Visual Design Media: Cartoons, Drawings and Photographs

Cartoons, drawings and photographs are frequently used to advertise (especially) unhealthy foods to children. Familiar media character branding, for example, is especially used for energy dense and nutrient-poor foods such as cookies, candy or chocolate compared to fruits or vegetables. The following paragraphs explain the effectiveness of the different design media on people’s food choices. Substantial research has investigated the impact of cartoons on children’s unhealthy food choices. Limited research, however, has focused on food drawings and the difference in effectiveness between food drawings versus photographs.

##### The Effect of Visual Design Media on Product Perceptions and Attitudes

The research of Lagomarsino and Suggs [227] examined the effects of cartoons versus drawings and photographs as a visual communication strategy on children’s healthy food choices. The children in the study **liked** foods displayed as cartoons the most, not the drawn foods, but they wanted to consume the foods represented by photos. These results show that when we want to motivate healthy food consumption by children, it may be more effective to use photos rather than cartoons or other animations. Nonetheless, research of Kraak and Story [228] suggest that cartoon media character branding compared with no character brand can positively increase children’s fruit or vegetable intake. So cartoons can still be effective in promoting healthy food. Furthermore, matching illustrations (vs. photographs) with organic food (vs. conventional) food increases advertising effectiveness [229]. In addition, matching illustrations (vs. photographs) with altruistic (vs. egoistic) claims can increase likelihood of purchasing and willingness to pay for the organic product [229]. Finally, Lazard et al. [230] show that the effects of photo manipulation (i.e., hue, saturation, brightness, and image content were enhanced beyond reality) as a visual cue in food marketing increased people’ perception of healthfulness, positive attitudes and purchase intentions. Also Peng and Jemmott [114] show that aesthetic appeal and specific photographic visual features influence the popularity of the food. It is unclear when cartoons, drawings and photographs can be used to promote positive attitudes towards healthy food (FR 31). Can they lead to higher **taste or quality** perceptions for all products (FR 32)? Do drawings or photographs increase **healthy, sustainability or size** perceptions?

##### The Effect of Visual Design Media on Other Psychological Processes

Septianto et al. [229] argue that illustrations are more effective than photographs to promote organic food because illustrations provide more abstract or less concrete information compared to photographs as they are a less realistic representation of reality. Similarly, organic food evokes more **abstract processing** than concrete processing as it increases the hypothetical distance of presence (only a minority of the food is organic). Do drawings also elicit more abstract processing? Do drawings (compared to photographs) facilitate our processing or lead towards an increase in recognition, but do photographs (compared to drawings) provide more information? Do cartoons, drawings or photographs influence **attention, affective, cognitive and motivational** reactions in a different way? When does this occur and does this affect food perceptions, attitudes and behavioural outcomes (FR 33)?

#### 4.3.4. Food Porn

Food styling is core to blogging and posting on social media such as Facebook, Instagram, Twitter, Snapchat, Pinterest, YouTube, etc. since it is useful in making the food look more appetizing and appealing to customers [231]. The term ‘food porn’ is increasingly used to describe the acts of excessive display, styling and capturing of food photographs on social media. These acts are added in order to elicit an invitation to gaze, to consume, and to tag images of food through digital platforms [232]. Petit et al. [233] state that the pleasure of viewing food on a screen now even exceeds the pleasure associated with seeing real food (particularly in the case of obese people). Research has primarily defined visual cues implied by food porn as erotic or based on the sexualisation of food [234]. Taylor and Keating [235] argue that recent exaggerated visual characteristics more accurately depict desires of human aspiration and reassurance. Despite its popularity, food porn on social media has attracted relatively less attention from the research community [236].

##### The Effect of Food Porn on Product Perceptions and Attitudes

Närvänen et al. [237] investigate the phenomenon of food porn in the context of food waste as social media enable also the aestheticisation of food waste. Their results show that the aestheticisation practices implied by food porn help move from negative meanings of food waste as something that looks and tastes bad towards more positive associations. Given that the research of Närvänen et al. [237] is—to the best of our knowledge—the sole paper investigating the causal effects of food porn on people’s attitudes and behaviour, further research on the effect of food porn on **food perceptions and attitudes** is required (FR 34). This paper also inspires future research on food porn in the context of healthy food consumptions and sustainable food consumption. Since Ibrahim [232] claims that food as a subject of digital photographs carries social resonance and sociability (i.e., social activity), food porn could be used to stimulate healthy or sustainable food consumption by setting a standard or social norm. Likewise, Petit et al. [233] hypothesize that food porn can make the consumption of healthy foods more appetizing and enjoyable (by allowing the brain to vividly imagine the consumption experience) and can lead to a better regulation of food intake.

##### The Effect of Food Porn on Other Psychological Processes

Taylor and Keating [235] examine key pictorial elements of food porn through the lens of visual arts. The authors suggest that creative disruption is a common strategy applied in food porn. This disruption creates a sense of intimacy and draws **attention** via activating desire-based triggers which engage the viewer. Although food porn has been normalized in contemporary food imaging, it does attract attention. The review of Spence et al. [238] of the growing body of cognitive neuroscience in the context of food porn demonstrates the effect that food porn images can have on neural activity, physiological and psychological response, visual attention (especially in the hungry brain). Looking at these pictures is known to trigger gustatory sensation in the brain and evokes a desire to consume [233]. Food porn elicits approach motivations and decreases self-control because of its excessive character. Further research could dig deeper into the effects of food porn on attention, **affective, cognitive and motivational** reactions and resulting effects on food perceptions, attitudes and behavioural outcomes. When do these effects occur? (FR 35)

### 4.4. Materiality

Materiality—although not defined by Adaval et al. [50] and less discussed in visual typologies—is an important visual design cue. Materiality does not only relate to touch as a sensory perception but also to sight. Sample et al. [7] define materiality as the visual texture and reactance of the exterior surface of an object as contained within the shape of that object. Materiality contains four important components: (1) ‘visual texture’ which is the apparent consistency of a perceived object’s surface; (2) reflectance, which is the propensity of an object to produce an image of the surrounding context on its surface; (3) opacity, that is the lack of transparency in an object’s surface; and (4) fluorescence, which refers to the propensity of an object’s surface to emit light through reflection or internal lighting [7]. Few research efforts (compared to other visual design cues) have investigated how materiality affects product perceptions, attitudes and behaviour [7].

#### 4.4.1. Visual Texture

Visual texture can contain, for example, the material a package is made of (e.g., plastic, cardboard, paper, thin foil; [66]).

##### The Effect of Visual Texture of Materiality on Product Perceptions and Attitudes

Mainly haptic reactions to packaging material are studied [24,239,240] as well as the sensory transfer between touch and flavour [30]. Some research looks at product expectations people have based on the package material. Rebollar et al. [207] found that paper packages of crisps were rated as higher **quality** and more artisan than metallic crisp packages, but this result could be due to an existing association between paper crisp packages and higher prices (as identified in a marketing analysis). Magnier et al. [66] found that products packaged in more (vs. less) sustainably-rated packaging (paper, cardboard vs. white plastic/aluminium) were rated as higher in quality. Paper packaging is associated with nature [66] and since natural materials are strongly associated with **healthiness [241]**, De Temmerman et al. [242] argue that paper packaging can also be associated with healthiness. Package design can create health halos causing people to think that the food is healthier than it actually is [38], which could increase choice. De Temmerman et al. [242] also found that a paper (vs. a plastic) package was rated more healthy and decreases unhealthy choices. They explain the effect of paper versus plastic packaging on food choice by the finding that healthy packages could activate a health goal leading to decreased unhealthy food choices [243]. If food is presented in a healthier perceived package, the consumer will be more likely to adhere to the long-term goal to eat healthily because self-control strategies are triggered [244,245]. According to Belei and colleagues [243], any cue in the marketing environment that makes the concept of health accessible in people’ minds and thus activates a health goal could lead to a decrease in unhealthy choices [243,246]. Interestingly, the finding that consumption decreased for food presented in paper packages disappeared when food was presented in a paper package with a ‘unhealthy’ association (i.e., a paper package resembling a typical French fries’ package) [242]. No research has been done on the relation between visual texture and **size** estimation. An interesting future research area could be to examine which, when and why visual texture factors affect food perceptions, attitudes and behavioural outcomes (FR 36).

Research also shows that people rely on material of packages to form judgments of packaging sustainability [66,73,95] but it is not clear how specific packaging materials can lead to different consumer responses. Steenis et al. [247] found that people rate glass jars and bioplastic pots as most **sustainable** (although these are not the most sustainable options according to the lifecycle analyses), but these sustainable ratings do not translate in more positive attitudes. Magnier et al. [66] showed that using sustainable material and showcasing it increases willingness to buy for highly (but not for low) ecologically conscious people.

Serving coffee in a smooth versus a rough ceramic cup makes the coffee **taste** more sweet and less acidic with a less dry aftertaste [248]. They did not find an effect of the haptic texture on aroma. Although this research concentrated on the haptic experience of cups, visual perception of materials could already suggest smooth or rough character of a container which could then evoke taste expectations by knowing which haptic experience to expect from a specific material. If a container looks rough on the outside, products could be rated as more crunchy or harder than when the container looks smooth at the outside [249]. Moreover, use of pliable materials like cardboard or paper could then decrease crunchy perceptions compared to non-pliable containers. This effect could depend on the product (FR 37). Piqueras-Fiszman and Spence [249] found no effects for yoghurt (for which crunchiness is not a key attribute). Also Slocombe et al. [250] did not find a significant effect of a rough versus smooth serving plate on the perception of sweetness, bitterness, and acidity of lemon curd although they did find an effect of the rough/smoothness coating of the food itself. Biggs et al. [239] did find that caramelized biscuits taken from a plate with a rough (vs. smooth and shiny) surface were rated as tasting crunchier and rougher while jellybeans were rated as chewier when served on the rough plate as compared to the smooth plate. Hence, the effect of package and plate material depends on the product itself. Carvalho et al. [248] argue that the rough feel of the plate could have primed the dominant textural property of the food itself, namely, crunchiness for biscuits and chewiness in the case of jelly babies. Other research also showed that rough (vs. smooth) bowls influenced the perceived saltiness of crisps, with ratings depending on the level of saltiness of the tasted crisps [251].

##### The Effect of Visual Texture of Materiality on Other Psychological Processes

No research to date has investigated the effects of visual texture on **attentional, cognitive, affective and motivational** reactions (FR 38). Could plastic or paper packaging evoke different levels of arousal or thoughts? This probably depends on the type of food and the perceived fit between the package and the food product (i.e., contrast with expectations and experience). Could plastic visual texture elicit more avoidance reactions than paper visual texture?

#### 4.4.2. Reflectance

##### The Effect of Reflectance of Materiality on Product Perceptions and Attitudes

The higher the glossiness (i.e., related to luminance [252]) of the product, the higher the perceived freshness of perishable food [253,254]. In a non-food context, glossy packages are also seen as more sophisticated [255], luxurious [256] and higher in **quality** [257]. Also De Kerpel et al. [258] found that food products in glossy packages are rated as less **tasty** and lower in quality and price. They did not find a difference in naturalness. They argue that glossy packages are seen as a signal of fatness. Therefore, a food in a glossy package is seen as less **healthy** (but no difference in sugar level inferences are found). These results suggest that glossy (vs. matte) food packages mainly serve as a signal of negative product qualities and even deteriorate actual taste experiences [258]. Ye et al. [259] found opposite results: although they did confirm that glossiness reduces healthiness perceptions, they found that glossy packages enhance expectations about tastiness. They explain this by learned associations: people are repeatedly exposed to glossy food packages containing unhealthy products and matte packages containing healthy products. Hence, associations between tastiness and healthiness of foods packaged in glossy versus matte packages are created. This is in line with Marckhgott and Kamleitner [260] who also found that less glossy packages were associated with perceptions of naturalness. However, the latter did not find that a better taste was associated with matte packages. However, De Kerpel et al. [258] argue that not only greasy foods but also foods with a high level of sugar are wrapped in those glossy packages so learned associations should exist between glossy packaging and sweetness. The latter could not be confirmed in their studies. They argue that the link between glossiness and greasiness cannot only be explained by learned associations but also by the fact that a food product’s surface signals the material of the package and quality of the food [261,262]. They further argue that glossy package serves as a cue to evaluate the food that is inside. A glossy surface may remind people of grease because fat and glossiness share some exterior resemblance. This could explain why they did not find an effect of the glossiness of the package on inferred sugar levels since glossiness and sugar do not share external characteristics. However, some questions remain (FR 39; FR 40).

In a non-food context, glossy (non-food) products (who are perceived as more wet than matte packaging, [263]) are generally preferred over matte products presumably by an innate preference for water [264]. In a food context, Han [265] showed that matte packaging was preferred over glossy packaging (for chocolate and granola). Marckhgott and Kamleitner [260] found that the link between glossiness and positive **product attitudes** does not occur for foods perceived to be artificial: for those foods a glossy package decreased positive attitudes since it decreased naturalness perceptions (e.g., ketchup and soda). Han [265] found that glossy packages induced lower purchase intentions because a glossy package induces a lower sense of trustworthiness especially for brands that do not signal high quality or premium value. People infer from glossy packaging that it is a tactic that marketers use to grab attention. People also ate more food from matte versus glossy packages [259]. The effects of glossy versus matte packaging seem, however, to depend on the product (FR 41). Further research could dig deeper into when and why glossy packaging induces positive food perceptions, attitudes and behavioural outcomes. For example, does healthiness, naturalness or perishability of the food inside the glossy or matte package play a role when making food choices? Does inherent glossiness of food (e.g., tomato vs. banana) play a role?

##### The Effect of Reflectance of Materiality on Other Psychological Processes

Limited research investigated the effects of reflectance on other psychological processes besides product perceptions and attitudes. Glossy objects attract **attention** [266] because they are aesthetically appealing [264] and/or because they serve as an information cue (e.g., help estimate three dimensional (3D) shapes of objects; [258]). People report an enhanced immediate attention (but not long-term attention) for glossy versus matte packaging [265]. People respond to glossy (shiny) things as almost as automatically as they do to lights [267]. It is unclear what the effect is of reflectance on **cognitive, affective and motivational** reactions and whether they can explain food perceptions, attitudes and behavioural outcomes (FR 42). It is possible that glossy packages are rated on their glossy character, while matte packages elicit other/more thoughts unrelated to the (lack of) glossiness of the package. Glossiness could be related to a lack of self-control since it can signal luxury or is associated with unhealthiness. Also glossiness could elicit a state of arousal as bright objects can evoke arousal [148,149].

#### 4.4.3. Opacity

##### The Effect of Opacity of Materiality on Product Perceptions and Attitudes

Billeter et al. [268] found that transparency in packaging increased trustworthiness, preference and purchase intentions when the product was visually appealing. Similarly, Chadran et al. [269] found in a non-food context that transparent packaging enhanced **quality** perceptions and willingness to pay for unknown products because product trust increased but known products were rated of less quality when packaging was transparent. Adding transparency can have a symbolic value as ‘transparency’ is synonymous with ‘openness’, ‘trustworthiness’, and ‘comprehensibility’ and transparency may promote representations of ‘being able to see (understand)’ something [270]. They further argue that a brand that shows openly what is inside package has got nothing to hide and could lead to increased trustworthiness. On the contrary, curiosity towards the product is higher for opaque packaging. People can anticipate or be intrigued by the content of the opaque package, increasing the value of the product when people can unveil it [271]. Vilnai-Yavetz and Koren [272] found that transparent packaging in case of a ready meal resulted in decreased sales, decrease quality ratings and aesthetics even though transparent packaging is rated as a very easy to use. Some questions remain (FR 43–FR 44). For example, does the effect of transparency on perceived food quality depend on the type of material (glass vs. plastic)? Why does this effect occur?

Sioutis [273] found that transparent packaging was rated more **healthy** than non-transparent packaging. Similar to Riley et al. [270], he found that transparency is rated important to judge healthiness since it gives people more information on the content of the package. Simmonds and Spence [257] also argue that a leaf-shaped window may instill the notation that the product is natural or fresh. They also suggest that a transparent window can signal a premium offering, innovativeness and modernity, freshness, honesty and quality assurance. On the other hand, previous research by Morales and Fitzsimons [274] seems to suggest that transparent windows or packaging would have negative effects of products rated disgusting and could even decrease **positive attitudes** of products nearby this package on the shelf or in the shopping cart (e.g., through contagion). We can question whether transparent packaging also has positive effects on food **sustainable** perceptions or food **size** estimations (FR 45). Seeing the content of the package could distort size estimations. Furthermore, where should a transparent window be placed to optimally affect product perceptions, attitudes and behavioural outcomes (FR 46)? Simmonds et al. [171] show that people have a general preference and display higher expected product tastiness for windows on the right-hand side. For example, in granola packaging, heavier granola will not appear in higher-placed transparent windows, possibly affecting estimations of variety of content while lower-placed transparent windows possibly show broken pieces possibly affecting quality perceptions.

##### The Effect of Opacity of Materiality on Other Psychological Processes

It is again unclear what effects are of transparency on other psychological processes besides perceptions and attitudes (FR 47). Does transparent packaging elicit concrete thoughts as concrete food features are noticeable? Is arousal lower since people know already what the package contains (hence, have less curiosity). Does it elicit approach behaviour?

#### 4.4.4. Fluorescence

Fluorescent materials do not produce their own light but reflect a different wavelength than received such that the surface appears to glow [275]. People are engaged with fluorescent products [276,277]. No research to date has investigated the effect of fluorescence of food, food packaging, in-store material or online store materials on psychological processes and behavioural outcomes (FR 48). For example, are people also engaged with fluorescent food? Are health perceptions of fluorescent packaging lower?

### 4.5. Text and Picture Combinations

Visual information can be provided in pictorial or verbal (when written) form or via a combination of pictorial and verbal information. At the point of purchase, packaging, information in offline or online ads but also other point-of-purchase (POP) material like price tags, promotional material etc. often contains both pictorial and verbal information. Little research, however, has addressed the effects when text and pictures are combined in visual presentations [278]. The specific combination of text and picture is what is called the relative picture word ratio [16]. The lack of research on the relative picture word ratio is surprising because so much of what people read is accompanied by pictorial information. Two notable exceptions (in a non-food context) are the studies by Carroll et al. [279] and Hegarty [280].

#### 4.5.1. The Effect of Text and Picture Combinations on Product Perceptions and Attitudes

No research to date has investigated what the effects of a specific picture–word ratio is on product **perceptions, attitudes** and behaviour. Research of Roose et al. [281] seems to suggest that visual cues in particular should be used to attract low involvement customers. Via a content analysis of food advertising, the authors found that informational cues (i.e., mostly verbal cues) are mostly prevalent in healthy food advertising while, transformational cues (i.e., mostly visual cues) dominate unhealthy food advertising. They argue that appealing to transformational cues would be more effective in the case of healthy food. Further research on the prevalence of informational cues and transformational cues is necessary in the context of healthy food packaging and how this impacts consumer decision making and how food involvement can moderate these effects (FR 49). Next, we ask what is the ideal amount of text to combine with a picture and whether this depends on the type of food (e.g., healthy vs. unhealthy food; processed vs. unprocessed food; sustainable food)?

#### 4.5.2. The Effect of Text and Picture Combinations on Other Psychological Processes

Most research on text and picture combinations in a non-food context focuses on which aspect of the ad captures **attention [282]** or on the impact of the size of the text and/or picture in the advertisement on viewers’ observation time. Carroll et al. [280] examined how people look at cartoons consisting of a single picture and a relevant caption (text) in a non-food context. People looked longer at the text than the cartoon and the cartoon was not given full inspection until the text had been read. Also, Hegarty [281] shows that the comprehension process is largely text directed. Similarly, Rayner et al. [278] found that viewers tended to read the large print, then the smaller print, and then they looked at the picture. Although people in general indicate that they prefer pictorial information over verbal information, they still spend a fair amount of time on the text in each ad (70% of the time they watched the ad) and read the text before carrying out a careful scan of the picture [278]. This would suggest that when more (or less) words are present (together with a picture), more attention would be given to this POP material and more thoughts could be evoked which could translate in higher purchase intentions. On the other hand, verbal information takes more time to process than pictorial information and viewers remember better pictorial aspects than textual aspects (namely superiority of memory for pictures over memory of words [283]) which would suggest that effective POP material should contain more pictures than words. We propose to investigate the effect of specific word-picture ratio’s on **attention, affective, cognitive and motivational** reactions and how they can explain downstream effects. We could ask, for example, whether text would receive the same attention in case of healthy or unhealthy food in-store ads (FR 50) or whether specific word-picture ratio’s decrease ease of processing and consequently increase arousal [284] and decrease self-control [285].

## 5. Spatial Processed Cues

Next to object processing, people also employ spatial processing. That is, they focus on the understanding where the object is relative to the self and its movement and transformation (50). Therefore, the three spatial-processed visual design cues described by Adaval et al. [50] are: (1) location, (2) movement and (3) the spatial relation between the object and the self. All three visual design cues reoccur in food literature, although to a different extent. In what follows the most important research in food on each of these three cues will be discussed and future research opportunities will be highlighted.

### 5.1. Location

Adaval et al. [50] names ‘location’ as the first visual design cue to be spatially processed. When observing location, viewers focus on distances, relative dimensions, and velocities [50]. Sample et al. [7] (p. 410) defines location as: “*The position, orientation, spacing, and movement of an object in relation to other objects in the area*”. In the following paragraph we describe the prevalence of position, orientation and spacing in food literature. As ‘movement’ is defined by Adaval et al. [50] as a main component of spatially processed design cues, we will discuss this cue not as a subcomponent of positioning, but as a main component in itself.

#### 5.1.1. Positioning

More specifically, the positioning is the placement of a figure within the background or in relation to another object [7]. For (food) marketers important questions to ask are: How to position your logo, claim or label on your product packaging? How to position displays or webpages (e.g., vertical or horizontal, left or right, etc.)? Is positioning able to induce both short-term and long-term effects regarding consumer choices (e.g., does a relative high logo positioning evoke associations of product superiority on the long term and in different contexts?)

##### Verticality in Positioning

The first dimension of positioning we discuss is the vertical dimension. That is the relative low or high positioning of a stimulus (e.g., claim, label, product images, product) against a certain background (e.g., package facade, display, webpage background, etc.). Research shows that verticality matters both in package design and product design. Vertical orientation influences both perceptions, attitudes and behaviour. Limited research, however, reports on the underlying processes.

(1) The effect of Verticality of Positioning on Product Perceptions and Attitudes

Machiels and Orth [286] show that people perceive a product (i.e., wine bottle) as more powerful when the label on the package is placed in a higher (vs. lower) vertical positioning. Similarly, Dong and Gleim [287] conclude that products positioning the logo higher on the package are likely to have more favourable perceptions due to perceived **quality** and this regardless of brand familiarity. In contrast, placing an image of the food itself at the bottom (vs. top) of the package façade enhances people’s flavour expectations [62]. Future research could investigate whether vertical positioning of labels, verbal claims or food pictures affect **taste, health**, **sustainable or size** perceptions and **attitudes** and behavioural outcomes (FR 51). Does a label on top of the bottle elicit product perceptions of luxury or exclusivity? Does this depend on the type of food so that, for example, healthy foods would be evaluated better when logos are placed at the top while centred logos would be better for unhealthy foods? Do food size estimations differ depending on the label placement?

(2) The Effect of Verticality of Positioning on Other Psychological Processes

Graham and Jeffery [288] and Cabrera et al. [68] show that label components at the top of the label were viewed more than those at the bottom. In addition, according to Cabrera et al. [68] nutrition facts labels with a central location on the label received more **attention** than those located peripherally. It is possible that the ideal label positioning depends on the content of the label (e.g., nutritional facts, sustainable character) (FR 52). Furthermore, Graham and Jeffery [288] argue that the most important label components could shift which would imply that the exact positioning of specific labels should be re-evaluated. For example, labels containing information on a nutrient like sugar merit a higher label position as sugar content has increased as a defining cue of food decisions [289]. Putting a sustainable label on top of a food package could attract attention to this label, increasing the chances that the sustainable character of the product will be used as a choice cue. Also within a nutritional label, the sugar content could be placed more on top (currently at the 4–10th position from the top of nutrition sheets, depending on the country/product category). Placing relevant information on top of a nutritional label could evoke more positive attitudes (since people can quickly find the information they are looking for) but can also attract attention to this informational cues making it more important in further choices (FR 53).

##### Horizontality or Laterality of Positioning

The second dimension of positioning is the horizontal dimension or the lateral positioning of package facade elements. Research shows that left versus right positioning matters in different contexts.

(3) The Effect of Horizontality or Laterality of Positioning on Product Perceptions and Attitudes

No research is found on the effects of horizontality or laterality in positioning on product perceptions and attitudes in the context of labels on product packaging. Future research could investigate whether horizontal positioning of labels on food packages affects product perceptions and attitudes (FR 54). 

(4) The Effect of Horizontality or Laterality of Positioning on Other Psychological Processes

Research on laterality of pictorial and textual elements on packaging shows that in order to receive the most direct attention, textual elements should be on the left-hand side of a food package, whereas pictorial elements should be on the right-hand side [290]. Other researchers found the opposite: labels positioned in the centre of the screen were viewed by a higher percentage of viewers and for a considerably longer time than those located at the sides [288]. It is possible that the content of the label (textual or pictorial) dictates the optimal position (FR 55).

#### 5.1.2. Orientation

Orientation can refer to the direction towards which the object is pointing (e.g., upwards vs. downwards, left vs. right, etc.) or to the angle of perception of an object or background, which is often called ‘camera angle’ (e.g., low angle shot, high angle shot, eye-level shot, bird-eye-level shot, etc.). Larsen et al. [53] describe this as one of the core components in video (marketing). But also in (in-store) advertising literature camera angle and its impact on consumer processing and decision making is often discussed [111,291,292]. Similar as to ‘positioning’ the orientation of a logo, label etc., camera angle can be described along a vertical dimension and a horizontal or lateral dimension. Orientation can also imply orientation of the product or stimulus against its background or orientation in direct relation to its observer. Given the importance of this product–viewer relationship a separate paragraph will be dedicated to this subject (see Section 5.3: Spatial Relation between Object and Self).

##### Direction

(1) The Effect of Direction of Orientation on Product Perceptions and Attitudes

Orientation of the product package refers to whether the package is mainly oriented horizontal or vertical (with the shape and positioning remaining constant). In line with the vertical-horizontal illusion [293], vertical dominant orientation of the package should make the package seem longer and narrower compared to a horizontal dominant orientation of the package. For the effects of products and packaging differing in height we refer to packaging dimensionality (see Section 4.2.1)

(2) The Effect of Direction of Orientation on Other Psychological Processes

Research on the downward-pointing triangle superiority (DPTS) effect shows that searching for a downward pointing triangle among upward-pointing distractor triangles goes faster than vice versa [294] since downward-pointing triangles are more likely to convey threat-related information than are upward pointing triangles. The former resembles angry faces in which the muscles are pulling down to form a ‘V’ shape and, therefore, captures **attention** more readily than do neutral stimuli such as upward-pointing triangles [295,296]. This could also imply that they evoke specific emotions like fear or induce self-protection which could lead to increased loss aversion or decreased risk-seeking [44]. Although downward and upward pointing triangles are rated as equally pleasant, Zhao et al. [297] found that choosing a bottle with a downward-pointing triangle on its label was faster than when the bottle presented an upward-pointing triangle on the label. The same results have been found for non-threatening images of triangular-shaped foods and pizza packaging [28].

Considering the global precedence of visual perception, Zhao et al. [297] wonder whether it is a possibility that the global outline shape of the stimuli (e.g., the downward-pointing triangle which could be threat-related) might be processed before the meaning of the stimuli (e.g., food or food packaging which are not threat-related). Future research is required on the order of visually processing shape (i.e., design) before meaning (i.e., content) and whether this depends on stimulus type (FR 56). For example, in the case of unhealthy food products would the orientation effect be stronger (as these can perhaps imply some connotations of danger regarding our healthy conditions) than in the case of healthy foods?

##### Orientation of the Background against Stimulus

Van Rompay et al. [298] show that vertical orientation of the background of a display (i.e., vertical stripes vs. horizontal stripes) not only affects impressions of product luxury, but also increases actual **taste** evaluations, including perceptions of taste strength (intensity), taste liking and price expectations. This effect of verticality on luxury perceptions does not always hold for food products. Van Rompay et al. [292] show that verticality cues should be used with respect to products that are naturally associated with luxury and related construct only. Thus, commodities (e.g., rice) or products which were never intended to represent power or luxury (e.g., low-budget products or services positioned as affordable and home-like) are unlikely to benefit from vertical cues signalling brand luxury and a high price. We wonder whether food products known for being sustainable would benefit from vertical cues as sustainability perhaps represents a ‘higher order’ status in a similar way as ‘luxury’ products do (FR 57)?

##### Orientation of Stimulus against Background

Elder and Krishna [298] explored the horizontal or lateral orientation of a product against its background. They show that viewers are more inclined to imagine drinking from a cup of coffee when the handle is orientated right than left against its background. That is, when the orientation of the handle of the cup matches the right handedness of the viewer. We wonder whether these effects would hold both in case the presented food is perceived by the viewer as tasty or untasty (FR 58)?

##### Camera Angle

(1) The Effect of Camera Angle on Product Perceptions and Attitudes

Van Rompay et al. [296] follow the embodied cognition framework suggesting that top view (vs. eye-level) and vertical orientation (vs. horizontal) background lead towards higher luxury and power perceptions. The effect of verticality on luxury perceptions is explained as following: because being physically (i.e., bodily) higher inspires feelings of power, one tends to attribute power-related meanings such as exclusivity and luxury to products (visually) associated with verticality cues. Furthermore, vertically downward or top shots negatively affect product **attitudes** and recall [113,299,300,301]. Research in a non-food context suggest that upward angles would increase size evaluations [299] and competence [300]. This would imply that vertically upward angles would increase **size** and **quality** perceptions of food (FR 59). What is the effect on other **product perceptions**? Can camera angle evoke perceptions of ‘competence’, ‘high value’ or even luxury in the case of sustainable products? Meersemann et al. [302] argue that people are less familiar with pictures of food in top (vs. diners’ eye) perspective which would diminish unhealthy choices through decreased food vividness and subsequently lowers people’ need for instant gratification. Further research could investigate how upward versus downward versus diners’ eye perspective could for example trigger healthy food choices. Would feelings of superiority evoked by a top-down perspective encourage people to choose more healthy foods?

Specific attention could be given to horizontal and vertical positioning of mobile screens. Since online shopping can occur on both a horizontal (such as when the tablet is rested on a table or on the knees) or vertical (when placed on an easel for example) position, it could be interesting to dig deeper into possible differences. Ardelet [303] show that mobile device tilt angle impacts purchase intention of advertised products. Purchase intention increases when device tilt angle enables the person to stand in a body posture that is consistent with the image perspective of the advertisement or webpage. In this case, it is more easy to mentally simulate. Since Ardelet [303] investigate mobile device tilt angle only in case ads were represented, we call for research to test these effects in an online shopping context. What about, for example, the effects of mobile device tilt angle when having an online consumer experience in an online (virtual) food shop?

(2) The Effect of Camera Angle on Other Psychological Processes

It is possible that different camera angles imply different distance estimations from the product on the picture. Downward angles could get the feeling that they are distant from the food (namely verticality theory [304]). On the other hand, looking down could be associated with proximity [293,305]. This could imply different **cognitive, affective and motivational** reactions like abstract or concrete processing, focus on different food attributes (feasibility/desirability) and more or less self-control (FR 60). It is possible that these effects are different for different product categories. Mulier et al. [306] show that vertical (vs. horizontal) video ads increase interest and increases processing fluency due to a reduced effort of watching this video on a smartphone (and not having to turn the smartphone to have an optimal viewing experience). Tilt angle could also affect other cognitive processes like abstract (for vertically oriented online shops) and concrete (for horizontal oriental online shops) processing and result in different behavioural outcomes.

#### 5.1.3. Spacing

Spacing refers to the distance between an intended focal object and additional information. A common example of spacing in marketing is the use of ‘white space’ or ‘negative space’. That is: the negative space or the conspicuously open space found between objects or other design elements within the borders of an (in-store) ad or a webpage [307].

##### The Effect of Spacing on Product Perceptions and Attitudes

No research to date investigated spacing in a food choice context. Based on research on spacing in ads that found that white space has proven to be effective to convey elegance, power, leadership, honesty, trustworthiness, a modern nature and a refined taste associated with the upper social strata [307], we could expect that food advertising could also positively affect perceptions of food **quality** and **taste**. **Size** estimations could become bigger when spacing between foods in advertisements increase. We could also ask what is an ideal spacing between label components or between package surface elements (FR 61).

##### The Effect of Spacing on Other Psychological Processes

We did not find any literature on the effect of spacing on other psychological processes in the context of product packaging and/or advertising. future research could investigate whether spacing between package surface elements affects attention, affective, cognitive or motivational reactions (FR 62)

### 5.2. Movement

Movement refers to a change in location of an object [50]. To derive movement the visual stimulus does not only need to be perceived along the dimension of distance or space, but also along the dimension of time [308]. Therefore, visual cues can imply information about the duration, movement, speed, and acceleration of a stimulus [16]. Movement as a visual design cue is a popular appeal today, especially in online food marketing. Weber et al. [309] show that almost all of the websites (88%) of the top five brands in eight food and beverage categories contained motion and movement as visual design cues. Despite this fact, research on movement or motion is scarce in the food marketing literature.

#### 5.2.1. The Effect of Movement on Product Perceptions and Attitudes

Awad et al. [310] show that depiction of food with motion either through live viewing or video lead to enhanced evaluations of both food freshness and appeal in multiple food domains, categories, and texture. Spence [311], on the other hand, found that moving food plates are seen as aversive and even ‘horrifying’ possibly due to our innate fear of eating food that has the capacity to move through its own decision or a taboo around harming living things. Mulier et al. [312] found that people have more positive **attitudes** of receding versus approaching stimuli. It is unclear whether approaching food items would be estimated as larger than receding food items or whether **food perceptions**, such as taste quality, would increase for receding stimuli (since it can increase loss aversion [312]) (FR 63).

Next to real motion—as in a change in location of an object—also static visuals can represent ‘motion’ [313]. That is, what is called ‘implied motion’. It broadly refers to the dynamic information extracted from static stimuli or pictures [314]. Within the context of food marketing ‘Implied motion’ is crucial [315]. An example of implied motion is a photograph that shows juice being poured into a glass, whereas a static photograph shows the same juice motionless in a glass. People perceive both solid, liquid, natural and artificial food presented with implied motion as more fresh and, therefore, also as more tastier than food presented without implied emotion [32,316]. Even static fruit presented in moving water was rated as more fresh [315].

A third way to represent ‘motion’ is what is called by Park and Rhee [317] ‘cinemagraph image’ in online trade contexts. The ‘cinemagraph’ part refers to the image with a special characteristic that provides the motion cue by playing the part of the picture endlessly in the form of a video [318]. Because the cinemagraph image is produced by the form of the GIF type in general, it is possible to apply to websites or most stores. Research on the effects of GIFs is scarce [319]. Mulier et al. [318] found in a non-food context that GIFs induce a sense of urgency which leads to positive brand attitudes. Do these effects apply in a food choice context (FR 64)?

#### 5.2.2. The Effect of Movement on Other Psychological Processes

Research, for example, shows that on-screen approaching food stimuli capture **attention** [320] whereas receding stimuli do not [321]. Considering these receding and approaching movements research of Hsee, et al. [322] reveal that people have an innate tendency of approach aversion. This means that approaching stimuli elicit more negative emotional reactions than non-approaching stimuli. More specific, approaching stimuli increase proximity and thus are potentially more harmful than non-approaching stimuli [322]. Hsee et al. [322] show that approach aversion has become overgeneralized. It occurs for stimuli that are both a priori negative and non-negative or even ambivalent, and for various types of stimuli. In addition, Mulier et al. [318] show that online advertisements using receding versus static products can increase positive feelings and that these effects are driven by a sense of loss. Future research could test whether these effects of approaching and receding stimuli are applicable in a food context and whether they differ between product categories. Approaching unappealing food could evoke different reactions than approaching appealing food (FR 65). We could argue that approaching (receding) food stimuli could also evoke different types of processing based on the distance they imply (i.e., construal level). What is the effect of the speed of the food stimulus? For example, a slowly receding food stimulus might evoke more positive feelings than a fast receding stimulus, which could alter the effects found in this research. Future research could also test what the underlying processes of implied motion are. Does it attract attention? Are the effects of approaching and receding movements also present when the movement is implied instead of actual movement?

Research on GIFs implies that people would attend to different cues. People who watched the GIFs attended more to price information [321]. The results of Mulier et al. [318] also imply that GIF type images could increase arousal. Further research is necessary to investigate how food GIFs elicit different **psychological processes** compared to static food images and whether this affects behavioural outcomes (FR 66). For example, does the repetition of the recurring movement, for example, facilitate memory in the long and/or the short term?

### 5.3. Spatial Relation between Object and Self

Next to being positioned or oriented low versus high or leftwards versus rightwards with respect to a certain background or other products, products can also be directly oriented in relation towards the viewer. Whereas positioning and orientation are merely describing the product and its background, the spatial relation between the object and the self focuses on the visual relation between the object and the viewer. The spatial relation between the object and the self is defined as the ‘perspective’ from which we see the stimulus [50].

First, the stimulus can be positioned with respect to the viewer as if the viewer is using or consuming the stimulus (i.e., first person perspective) or as if somebody else does (i.e., third person perspective). The third-person imagery perspective occurs when an individual sees himself in the image from the visual perspective that an external observer would have, looking in from the outside, whereas the first-person imagery perspective occurs when the individual adopts a natural visual perspective, looking out at the situation through his own eyes [323,324,325]. Similarly, according to Saine et al. [325] when people imagine themselves in various consumption scenarios, they can do so from the actor perspective (i.e., first person) or the observer perspective (i.e., third person). These different vantage points are known as imagery perspectives. It is important to note that in both imagery perspectives the main character within each scenario is the same (i.e., the people view themselves in both events but from a different perspective). A second and previously discussed way in which products can spatially relate to the viewer is, for example, the vertical positioning (see Section Verticality in Positioning).

#### 5.3.1. The Effect of the Spatial Relation between Object and the Self on Product Perceptions and Attitudes

Saine et al. [325] show that third person (vs. first-person) simulation were characterized by (1) switching to an alternative instead of staying with the status quo (default option), (2) fewer sensory components, (3) worse **attitudes**, and (4) a decreased willingness to pay for tempting foods. Seeing products from a third versus first perspective could also shrink perceptions of objects, such as money and time [326,327], possibly leading to more negative **product perceptions** and behavioural outcomes (FR 67).

#### 5.3.2. The Effect of the Spatial Relation between Object and the Self on Other Psychological Processes

The research of Saine et al. [325] put forward that it is easier to simulate first (vs. third person) perspectives. Furthermore, the brain regions associated with reward processing are more active when imagining the taste of a tempting food compared to viewing it passively [328]. Given that a third-person point of view deemphasizes the sensations and neural activations caused by imagined eating experience, this form of simulation is less likely to produce the feelings of reward that heighten motivation to consume [328,329,330,331]. Saine et al. [325] point out that future neuroimaging studies will be instrumental in explicating how the activation of the so called ‘pleasure centres’ of the brain, in relation with those regions involved with self-control, influence the relationship between the visual perspective adopted during imaginary eating and actual consumption (FR 68). Also they call for research on the crucial role of third-person imagery as a buffer from the low-level sensations (e.g., imagined pleasure or pain) that lead people to engage in goal-incongruent behaviours. For example, could imagining eating less tasty vegetables from a third-person view reduce the thoughts regarding how distasteful it is, but increase the positive self-concept thoughts associated with eating healthily? We could also expect that a third- (first)-person perspective would increase the psychological distance one feels towards the food product which could evoke more abstract (concrete) processing [332] which could imply more self-control and healthy choices [332,333,334,335].

## 6. Further Considerations

As described above, specific visual design cues can affect psychological processes and behavioural outcomes. We provided examples of previous research and possible future research on the effects of visual design cues on attention, affective, cognitive and motivational reaction, product perceptions and attitudes which in turn can affect behavioural outcomes. After having reviewed the specific research opportunities, we now formulate some general recommendations for future research investigating effects of visual design cues.

First, to optimally understand the effect of visual design cues, one must be aware of the interaction between different visual design cues. Previous research shows, for example, that colour affects how shapes are perceived [74]. Basis forms such as pyramids and cubes could be perceived as more stable and solid when they are presented in a monochrome colour, while complicated forms would be seen as durable due to the monochrome colour. People prefer particular shapes to have specific colours (red for circle or squares, yellow for triangles, Albertazzi et al., [336]). Furthermore, a combination of different visual cues leads to different product inferences and liking/purchase intent [170,337]. Black has a stronger implicit association with unhealthfulness than red but the opposite trend holds in the context of warning signs [68]. A combination of angular/green or round/pink coffee labels received higher liking and purchase intent ratings than the angular/pink and the round/green combined ones [93]. No effects on health and taste perceptions were found. The findings show the importance of matching multiple packaging elements such as shape, colour, texture in a specific context. Ares and Deliza [207] argue that visual design cues of the package should be congruent with the expected texture, taste, and calorie intake of the food in order to generate positive experiences among people. Incongruent elements elicit ambiguity with respect to product identity and, therefore negatively impact product attitudes. Congruent elements, then, facilitate processing fluency and as such, also positive attitudes. Further research is required regarding which combination of visual design cues optimally affect behavioural outcomes, how these effects can be explained, and in which contexts (e.g., product types) these interaction effects are present or absent (FR 69).

Next, it is important to understand that people not only perceive visual signals, but we combine perceptions from different senses to interpret and react to environmental stimuli [165]. Vision interacts with input received from the haptic, auditory, olfactory and gustatory systems to make judgments [16]. Recently several authors focused on the effects of cross modal correspondence [170,338,339] for reviews see for example [340,341] or the association of information from one sensory feature with another sensory feature from a different sensory modality [166]. When sensory dimensions are combined, cross modal correspondences are acquired [166]. Cross modal correspondences may occur when the terms which are used to describe the stimuli in two sensory dimensions overlap (e.g., “low/high’: spatial and sound; “sharp/round”: visual and gustatory, [170]). Cross modal correspondence between different senses might be context-dependent (i.e., presentation mode, product type, etc.). Future research is needed to unravel potential moderators (FR 70).

Recently we have seen that the landscape of food choice and food choice research is changing due to the high pace of technological (r)evolutions. An increasing number of food choices do not take place in a mere offline setting. Today, we have online devices such as mobile apps which help people in offline settings such as the store environment via providing online information. Next, online to offline (O2O) settings are created in which the food can be searched for and bought online and consumed offline [342]. Virtual reality and augmented reality, then, try to emerge the consumer into a real-life experience via imagery evoking visual design cues while being actually 100% online. To summarise, consumption experiences today can take place on a continuum ranging from being 100% offline via hybrid forms to being 100% online. Although online shops—with or without the help of virtual reality (VR) and/or augmented reality (AR)—are trying and almost succeeding to induce the consumer into an experience which feels real, some differences with an offline setting cannot be taken away. Important in the context of food is that the online context due to its limited nature lacks touch and smell as informational cues for making food choices. Consequently, for digital marketing both communication and product presentation through visual perception is of major importance [343]. Benn et al. [344] showed that in an online grocery shopping context, participants looked at the pictures of food products, rather than examining detailed product information. Adaval et al. [50] state that in an online context visual cues are used increasingly to elicit mental imagery (e.g., [345]). Future research could investigate whether visual design cues have a different impact in an online or offline food store environment. Colour saturation of packaging for example, could be perceived differently online versus offline so that higher saturation levels should be used online in order to acquire the same positive effects on behavioural outcomes as in an offline context. Huyghe et al. [346] show that the symbolic representation of grocery products in an online shopping channel decreases products’ vividness implying that visual design cues will have different effects in an online versus an offline shopping channel (FR 71).

Furthermore, the O2O explosion has brought forward a growth in mobile communications [347]. Mobile applications are applied in all kind of food choice contexts such as restaurant, food delivery places and in-store environments. However, what factors influence people’s O2O food choices are still not well understood [342]. Given this popularity, it is important to investigate the effectiveness of visual design cues in mobile communications (FR 72). Dunford et al. [348] reveal, for example, that traffic light labeling was chosen as the preferred visual format for presenting nutritional information in a mobile app. Cho et al. [347] found that the importance of visual design cues depends on type of household, while Xu and Huang [342] show that, among other things, whether image cues influence people’s expectations depends on the level of need for cognition. These latter papers reveal the effect of various moderating variables (i.e., numbers of persons in households, need for cognition). Further research on these and other potential moderators (e.g., brand familiarity) is needed.

In the context of food marketing long-term effects of visual design cues have not been studied in a systematic way [348]. In terms of data-collection, however a growing stream of data becomes available as a result of the growing digitization of our economy, and an ever extending set of analytical tools is becoming available to deal with this type of data [349]. It could be interesting to see how, for example, increased exposure to specific visual design cues is capable to repeatedly decrease emotional disgust reactions and increase acceptance of unfamiliar or odd food products in the long run [183]. People who are triggered by specific visual design cues to consider particular food products could develop more interest in that particular food product [350] or learn more about that product [351] which could trigger repeated behaviour. Future research could investigate whether visual design cues that evoke affective (e.g., increased pleasure) or cognitive reactions (e.g., increased amount of thoughts) or both increase acceptance of these food products in the long run (FR 73).

People’s food choices are often driven by reasons of which people are not (fully) aware. The process of making food choices is influenced by a complex set of emotions, feelings, attitudes, and values which are impossible to assess simply by asking people their opinions [352]. Traditional techniques, such as self-reports or interviews, mainly allow measuring conscious and rational reactions to food products or food marketing strategies [352]. Recently, there has been increasing interest in the multidisciplinary field of ‘neuromarketing’, which adopts neuroscientific techniques like the measurement of facial expressions, heart rate variability, galvanic skin response and gaze deviation to study objectively and in a observable way consumer behaviour, such as emotional and spontaneous reactions [352,353,354]. These neurophysiological measures make it easier to examine also the level and focus of attention and neural activation [10]. In particular, in the case of visual design cue variables, such as attention and emotion, technologies such as eye-trackers, face readers, galvanic response, brain-scanning equipment and implicit measures are required. Pentus et al. [355] for example, show that combining conjoint analysis with psychophysiological measurements makes it possible to detect how important different visual design cues regarding food package are in generating positive emotions towards potential consumers. Also, Songa et al. [356] show that a combination of implicit and explicit measures of attitudes are necessary to explain people’s reactions towards sustainable logos. Mixed results exist on the predictive nature of implicit and explicit measures. Implicit attitudes, for example, have been found to influence the use of colour-coded information in choosing products while explicit attitudes were not predictive of behaviour [357]. Future research should use implicit as well as explicit measures to investigate psychological processes and behavioural outcomes and investigate the specific relation between implicit and explicit attitudes regarding visual design cues, and its impact on food choices (FR 74). The role of implicit attitudes is expected to be much stronger in food markets since they are characterized by significant time pressure and automaticity [358].

Throughout previous research, it became clear that the effect of visual design cues on food choices is often moderated by several factors. In the final paragraphs we want to provide a more structured and brief insights into which type of moderators are at play when visual design cues impact in-store food choices. Research [359] on moderating variables in food marketing makes the distinction between internal food factors (i.e., characteristics of the food product itself) and external food factors (i.e., individual differences and contextual factors) relevant. Future research on the effectiveness of visual design cues on food choices should consider whether these moderators could have an important role.

Foods can be categorized by different types of categorization and dimension: healthy versus unhealthy, processed versus unprocessed, sustainable versus unsustainable, familiar versus unfamiliar, authentic and old versus new, product value (e.g., functionality, quality and ease of use) [134], price etc. We provided ample examples above how a fit between the specific type of visual design cue and the specific characteristics of the food product is core to appeal to people. Therefore, we suggest future research consider intrinsic food product characteristics when investigating the effects of visual design cues on behavioural outcomes and psychological processes.

Besides internal food factors, also the context impacts the effect of visual design cues on food-decision processes. Some contexts, such as, busy shopping streets make it harder for visual design cues to draw attention. We also expect differences in the effects of visual design cues depending on the online or offline context (see above). Furthermore, specific type of shops (e.g., low budget, bio-food shops) could raise different expectations concerning the presence of visual design cues. Visual design cues need to match with the expectations that people have from the specific store. Therefore, we urge researchers to also consider the specificity of the food store context in which the visual design cues will be implemented. Also, the time of day, weather conditions or seasonal effects could alter people’s reactions to visual design cues. A bright saturated colour could brighten one’s rainy day while the same package colour could have less effect on a sunny day. In addition, people’s antecedent states could affect the effectiveness of visual design cues [1]. When people enter a store for example, they could already experience specific emotions, time pressure, etc. We could expect colour or shape preferences to change depending on how people feel when they enter the store/visit the website. Also, visual design cues will probably have a higher impact on food choices when decisions are made rapidly and under cognitive load (i.e., stressed, rushed, tired).

Lastly, there are several individual characteristics that affect the effectiveness of visual design cues on consumer choices. These personal factors include demographic variables like age and gender [1] and social economic status [360], psychological and physiological needs and traits [1] and food centeredness. The latter is manifested in different constructs like food involvement, dieting behaviour and restrained eating, binge eating and food neophobia. Food *involvement* (i.e., the level of importance of food in a person’s live, [361])*,* for example, has been emphasized as an important variable in food choice [362]. It is possible that the impact of visual design cues as marketing strategy on consumer decision making will be less strong for highly involved people compared to less involved people. Furthermore, research shows that the influence of visual design cues like package shape size depends on the level of restrained eating a person adheres [363,364]. In addition, *food neophobia* could moderate the effects of visual design cues on choice behaviour. We wonder whether in case people have high levels of food neophobia they favour traditional plates over contemporary plates as traditional designs are often what people are more used to. Also, visual design cues stimulating novel foods will be less effective for people high (versus low) in neophobia.

Several psychological traits like imagery as a trait, the way we process information (i.e., visual vs. verbal), self-control, health-locus of control, goals and motivations can also affect the influence of visual design cues on food choices. According to Raghubir and Krishna [365] people differ in their tendency to use mental pictures to solve problems and have different preferences towards processing visual versus verbal information. In general, imagery as a trait affects the extent to which individuals rely on visual stimuli to make choices [16]. When creating visual design cues it is important to be aware of this trait and to search for ways in which you are able to draw the attention and encourage the involvement of people who score high on imagery as a trait, but also for people who score low on this trait.

Related to imagery as a trait, is *the way we process information* as a trait. Some people called verbalizers prefer verbal or written information, while visualizers prefer visual information [366]. Further research could, for example, take place regarding which type of visual design for letters is more or less appealing and easy to process for verbal processers. Also, studies that test the effectiveness of specific properties in store visual displays should incorporate this individual difference variable.

*Self-control* as a psychological actor moderates food decisions too [367,368]. Visual design cues could have less effect on food consumption decisions for people with high (vs. low) food-related self-control. Next to self-control, also Health Locus of Control (HLOC) moderate consumption effects [369]. HLOC is defined as the degree to which a person believes his/her health is controlled by internal or external factors [370]. Individuals with an internal HLOC (i.e., health internals) believe that their health is related to their own behaviours, while those with an external HLOC (i.e., health externals) believe that their health is controlled not by themselves but by external factors such as fate, and powerful others. The level of HLOC could affect whether visual design cues are effective in stimulating healthy choices.

Last but not least, *goals and motivations* also have a strong impact on specific behavioural outcomes in a food choice context. Previous research shows individual factors like the shoppers’ overall trip goals or store-specific shopping objectives affect decisions in the choice context [359]. According to Raghubir [16], the goal one adheres to steers first fixations, and as such also the information that shoppers will process. People give attention to what is motivationally relevant and process information accordingly [371,372]. Goals can vary from physiological (e.g., hunger) or concrete (e.g., look for lowest price food product) to psychological (e.g., display power) and abstract goals (e.g., act sustainably). Goals may affect whether one pays attention towards the stimulus, but they may also affect attitudes. For example, when hungry, for example, people use information processing that is primarily influenced by basic and swift reactions regulated by the hedonic system and are especially open to hedonic and palatability cues [18]. The research of Shimizu et al. [373] shows that the effect of environmental cues on intake was most pronounced among participants who were hungry. Deng and Kahn [374] show how goals impact whether one prefers a light or heavy product thereby affecting the ideal positioning of an image on a package (i.e., high is light and low is heavy). Hence, taking into account the goals people have when making food choices could improve the effectiveness of visual design cues. For example, it could be more effective to show easy pictorial static information when the goal is to provide information while moving visual stimuli could attract more attention or provide aesthetic pleasure. Also, entering the store without a concrete goal (e.g., unplanned versus planned shopping) could make one more open to the persuasive effects of visual design cues. Further research should, then, consider the goals people have at the point of purchase, as these will probably affect the visual design cues people attend to and are persuaded by.

## 7. Conclusions

By identifying and underpinning a future research agenda, the present review aims to contribute to tackling the challenge of understanding how visual design cues can affect behavioural outcomes in a food choice context. We brought together research on the effect of object processed and spatially processed visual design cues on behavioural outcomes in a food context and discussed how they can be explained by several psychological processes including attention, cognitive, emotional and motivational reactions, product perceptions and attitudes. We advance future research by providing interesting research gaps to further investigate whether and why specific visual cues affect behavioural outcomes in a food choice context. We also provide some general recommendations taking into account the current in-store context (offline and online), the state-of-art in measuring psychological processes and behavioural outcomes, and the specific product-, person- and context-related moderators. With this review, we offer guidance for future research to untangle the complexity of the effect of visual design cues on behavioural outcomes in a food choice context.

## Figures and Tables

**Figure 1 foods-09-01495-f001:**
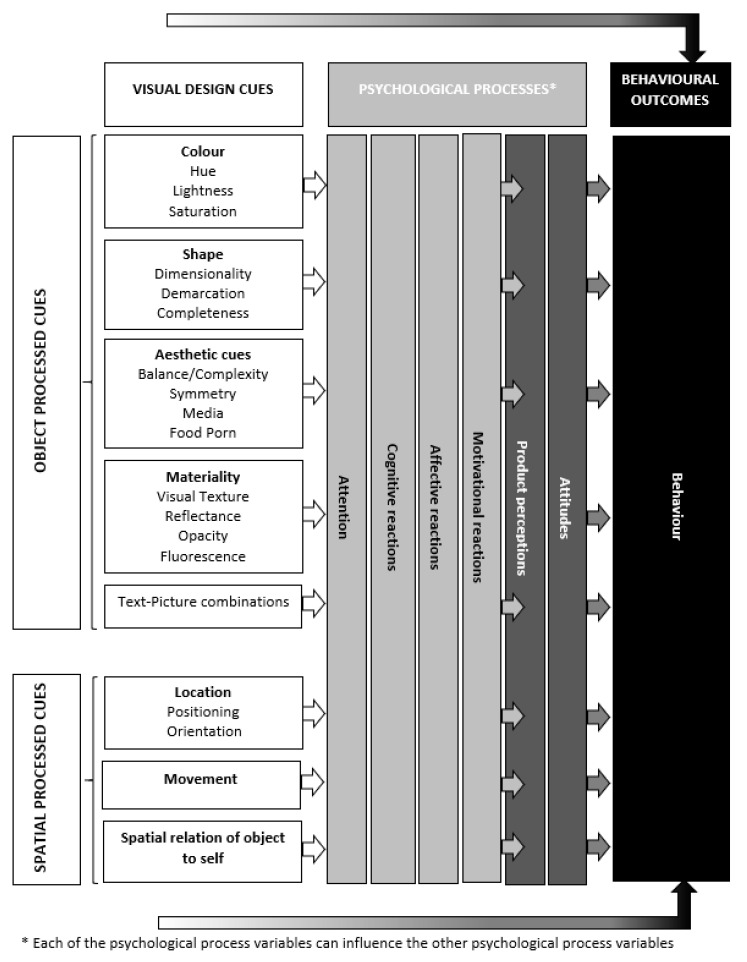
Overview of visual design cues, psychological processes and behavioural outcomes discussed in this paper (adapted from Sample et al. [7], Adaval et al. [50] and Labrecque et al. [39].

**Table 1 foods-09-01495-t001:** Overview of specific behavioural outcomes and psychological processes.

Psychological Processes	Examples of Specific Psychological Processes
Attention	Attention
Cognitive reactions	Amount of thoughts Abstract-concrete thoughts Divergent-convergent thinking Analytical skills Creativity Processing fluency Heuristic/elaborative processing Intuitive/deliberate decision making
Affective reactions	Valence Arousal Specific emotions (e.g., happiness, anger, fear)
Motivational reactions	Approach avoidance motivation Self-control Self-protection Status seeking Engagement to win?
Product perceptions	Quality perception Taste perception Health perception Sustainability perception Size perception
Attitudes	Attitudes
Behavioural outcomes	Examples of specific behavioural outcomes
Behaviour	Intentions Choice Willingness to pay

**Table 2 foods-09-01495-t002:** Visual design cues: our classification; classification of Adaval et al. [50] and Sample et al. [7].

CLASSIFCATION IN PAPER	SUB-DIMENSIONS IN PAPER (Number Heading)	Adaval et al.	Sample et al.
Object processed cues			
1. Colour	Hue (4.1.2) Lightness (4.1.3) Saturation (4.1.4)	Colour	Illuminance; surface colour
2. Shape	Dimensionality (4.2.1) Demarcation (4.2.2) Completeness (4.2.3)	Shape	Shape
3. Aesthetic cues	Balance/Complexity (4.3.1) Symmetry (4.3.2) Media (4.3.3) Food Porn (4.3.4)	Aesthetic	
4. Materiality	Visual Texture (4.4.1.) Reflectance (4.4.2) Opacity (4.4.3) Fluorescence (4.4.4)		Materiality
5. Text and picture combinations	Text and picture combinations (4.5)	Text picture and combinations	
Discussed throughout the paper		Logo	
Spatial processed cues			
6. Location	Positioning Verticality (5.1.1.1) Horizontality (5.1.1.2) Orientation Direction (5.1.2.1.) Background-Stimulus (5.1.2.2.) Stimulus- Background (5.1.2.3.) Camera Angle (5.1.2.4.) Spacing (5.1.3.)	Location	Location
7. Movement	Movement (5.2)	Movement	
8. Spatial relations of the object to the self	Spatial relation between object and self (5.3.)	Spatial relation of object to self	

**Table 3 foods-09-01495-t003:** Examples of visual design cues in a food context.

Visual Design Cue	Examples in Food Context
Colour	Package colour; Food colour; Label colour; Logo colour; Colour coding
Shape	Package shape; Food shape; Label shape; Logo shape
Aesthetic cues	Complex and balanced design; Symmetrical design, Cartoon, Drawing, Photograph, Food porn
Materiality	Food texture; Package texture; Food glossiness; Package glossiness; See-through vs. opaque packages; Fluorescent food; Fluorescent packaging
Text and picture combinations	Transformational cues vs. informational cues
Logo	Logo colour; Logo position; Logo movement
Location	Positioning of logos and labels; Food positioning on food shelf; Orientation of background against stimulus; orientation of stimulus against background; Camera angle; Positioning mobile screens; Mobile device tilt angle; Spacing
Movement	Moving Logos; Moving ads; Implied movement
Spatial relation between object and the self	Third versus first-person perspective

**Table 4 foods-09-01495-t004:** Table of contents of effects of visual design cues.

4. Object processed cues	
4.1. Colour	
4.1.1. Colour associations	
4.1.2. Hue	The effect of hue on product perceptions and attitudes The effect of hue on other psychological processes
4.1.3. Lightness	The effect of lightness on product perceptions and attitudes The effect of lightness on other psychological processes
4.1.4. Saturation	The effect of saturation on product perceptions and attitudes The effect of saturation on other psychological processes
4.2. Shape	
4.2.1. Dimensionality	The effect of dimensionality of shape on product perceptions and attitudes The effect of dimensionality of shape on other psychological processes
4.2.2. Demarcation	The effect of demarcation of shape on product perceptions and attitudes The effect of demarcation of shape on other psychological processes
4.2.3. Completeness	The effect of completeness of shape on product perceptions and attitudes The effect of completeness of shape on other psychological processes
4.3. Aesthetic cues	
4.3.1. Balance and complexity	The effect of balance and complexity on product perceptions and attitudes The effect of balance and complexity on other psychological processes
4.3.2. Symmetry	The effect of symmetry on product perceptions and attitudes The effect of symmetry on other psychological processes
4.3.3. Visual design media: cartoons, drawings and photographs	The effect of visual design media on product perceptions and attitudes The effect of visual design media on other psychological processes
4.3.4. Food porn	The effect of food porn on product perceptions and attitudes The effect of food porn on other psychological processes
4.4. Materiality	
4.4.1. Visual Texture	The effect of visual texture of materiality on product perceptions and attitudes The effect of visual texture of materiality on other psychological processes
4.4.2. Reflectance	The effect of reflectance of materiality on product perceptions and attitudes The effect of reflectance of materiality other psychological processes
4.4.3. Opacity	The effect of opacity of materiality on product perceptions and attitudes The effect of opacity of materiality on other psychological processes
4.4.4. Fluorescence	
4.5. Text and picture combinations	4.5.1. The effect of text and picture combinations on product perceptions and attitudes 4.5.2. The effect of text and picture combinations on other psychological processes
5. Spatial processed cues	
5.1. Location	
5.1.1. Positioning	Verticality in positioning (1) The effect of verticality in positioning on product perceptions and attitudes (2) The effect of verticality in positioning on other psychological processes Horizontality or laterality in positioning (1) The effect of horizontality and laterality in positioning on product perceptions and attitudes (2) The effect of horizontality and laterality in positioning on other psychological processes
5.1.2. Orientation	Direction (1) The effect of direction of orientation on product perceptions and attitudes (2) The effect of direction of orientation on other psychological processes Orientation of the of background against stimulus Orientation of stimulus against background Camera angle (1) The effect of camera angle on product perceptions and attitudes (2) The effect of camera angle on other psychological processes
5.1.3. Spacing	The effect of spacing on product perceptions and attitudes The effect of spacing on other psychological processes
5.2. Movement	5.2.1. The effect of movement on product perceptions and attitudes 5.2.2. The effect of movement on other psychological processes
5.3. Spatial relation between object and the self	5.3.1. The effect of the spatial relation between object and the self on product perceptions and attitudes 5.3.2. The effect of spatial relation between object and the self on other psychological processes

**Table 5 foods-09-01495-t005:** An overview of the suggested research possibilities.

	(Sub-) Dimension	Psychological Processes	Future Research Questions
**COLOUR**			
FR 1	Hue	Quality perception	How do people perceive and assess quality of food with ‘new’ hues? Does this affect behavioural outcomes?
FR 2	Hue	Taste perception	What taste inferences do people make for food products with unexpected hues? Does this affect behavioural outcomes?
FR 3	Hue	Health perception	Does label, package or logo hue affect health and nutritional value perception and hence behavioural outcomes? Does the type of food (e.g., healthy or unhealthy) or food hue moderate this effect?
FR 4	Hue	Sustainability perception; Size perception	Does label, product, package or logo hue affect sustainability or size perceptions of food products? Does this affect behavioural outcomes?
FR 5	Hue	Attitude	When and why are specific hues more liked than others and does this translate in behavioural outcomes?
FR 6	Hue	Attention; Cognitive reactions; Affective reactions; Motivational reactions	In which contexts and for which foods does colour coding stimulate healthy and unhealthy consumption? Can this be explained by attention, cognitive, affective or motivational reactions?
FR 7	Hue	Attention	Which hues and hue combinations attract most attention in which contexts? Does this affect food perceptions, attitudes and behavioural outcomes?
FR 8	Hue	Attention; Cognitive reactions	Do specific hues focus attention to specific food product features and which cognitive processes underlie these effects? What are the downstream effects on food perceptions, attitudes and behavioural outcomes?
FR 9	Hue	Cognitive reactions	Do specific hues influence processing difficulty, analytical skills, convergent/divergent thinking and other cognitive reactions? What are the effects on food perceptions, attitudes and behavioural outcomes?
FR 10	Hue	Affective reactions	What are the effects of product, packaging, label, logo or ambient hue on feelings and mood states and resulting food perceptions, attitudes and behavioural outcomes? Do these effects depend on the type of food product?
FR 11	Hue	Affective reactions	Do people have different affective reactions to black and white versus other hues? Do these affective reactions depend on the context? Does these affective reactions result in other food perceptions, attitudes and behavioural outcomes?
FR 12	Hue	Motivational reactions	Can specific package, product or logo hues evoke approach motivations and hence trigger positive behavioural outcomes for types of less liked sustainable food like insect based food, cultured meat, ugly looking food and vegetables? Does this effect depend on the level to which food appearance differs from the norm?
FR 13	Lightness	Taste perception; Sustainability perception; Size perception	What is the effect of colour lightness of food product, packaging, labels and logos on taste perception, sustainability and size perception? Does this affect behavioural outcomes?
FR 14	Lightness	Attention Cognitive reactions; Affective reactions; Motivational reactions	Can lightness of food product, packaging, label and logo colour affect attention, cognitive, affective and motivational reactions and which effects does this have on perceptions, attitudes and behavioural outcomes?
FR 15	Saturation	Food perceptions	What is the impact of food colour saturation on quality, taste, healthiness, sustainability and size perception? Is this different for different products (for example fresh produce, fruit juices, dairy drinks?) and for different ripeness levels of fresh produce? Does this affect behavioural outcomes?
FR 16	Saturation	Food perceptions	What is the impact of the saturation of the packaging colour on food perceptions like quality, taste, healthiness, sustainability or size perception? Are these effects different for different food categories? Does this affect behavioural outcomes?
FR 17	Saturation	Cognitive reactions; Motivational reactions	What is the effect of colour saturation on cognitive and motivational reactions? What are the effects on product perception, attitudes and behavioural outcomes?
**SHAPE**			
FR 18	Dimensionality	Food perceptions	Does shape dimensionality of the food product, package, label or logo affect taste, quality, healthiness or sustainable perceptions of the product or package? Under which conditions? Does this affect behavioural outcomes?
FR 19	Dimensionality	Attention; Cognitive reactions; Affective reactions; Motivational reactions	What is the effect of package height, width and length on attention, cognitive, affective and motivational reactions and how can they explain effects on food perceptions, attitudes and behavioural outcomes?
FR 20	Dimensionality	Quality perception	Does unusual or unique shaped food packaging enhance quality perceptions? When and why does this occur? Does this affect behavioural outcomes?
FR 21	Dimensionality	Taste perception	Does unusual or unique shaped food packaging enhance taste perceptions? When and why does this occur? Does this affect behavioural outcomes?
FR 22	Dimensionality	Food perceptions	How can negative perceptions towards unusual food shapes be countered?
FR 23	Demarcation	Food perceptions Attitudes	Can package demarcation trigger increased healthy or sustainable food consumption? How can these effects be explained?
FR 24	Demarcation	Attention; Cognitive reactions; Affective reactions; Motivational reactions	What is the effect of demarcation on attention, cognitive, affective and motivational reactions and how can they explain effects on food perceptions, attitudes and behavioural outcomes?
FR 25	Completeness	Food perceptions Attitude	What is the effect of food product incompleteness on quality, taste, healthy or sustainability perceptions and attitudes and does this affect behavioural outcomes?
FR 26	Completeness	Attention; Cognitive reactions; Affective reactions; Motivational reactions	What is the effect of food product incompleteness on attention, cognitive, affective and motivational reactions and how can they explain effects on food perceptions, attitudes and behavioural outcomes?
**AESTHETIC DESIGN**			
FR 27	Balance/ Complexity	Food perceptions Attitude	Which combination of balance and complexity of food presentation and packaging increases food perceptions, attitudes and behavioural outcomes. How can these effects be explained?
FR 28	Balance/ Complexity	Cognitive reactions; Affective reactions; Motivational reactions	Does balance and complexity of food presentation and packaging influence affective, cognitive and motivational reactions? When does this occur? Does this affect food perceptions, attitudes and behavioural outcomes?
FR 29	Symmetry	Food perceptions	What is the effect of symmetrical design on quality, taste, healthy, sustainable or size perceptions? Is this effect contingent on product category? Does this affect behavioural outcomes?
FR 30	Symmetry	Cognitive reactions; Affective reactions; Motivational reactions	Does food package and product symmetry influence affective, cognitive and motivational reactions? When does this occur? Does this affect food perceptions, attitudes and behavioural outcomes?
FR 31	Media	Food perceptions; Attitude	When can cartoons, drawings and photographs be used to promote healthy food? Why does this positive effect occur?
FR 32	Media	Food perceptions	Do cartoons, drawings or photographs increase or decrease taste, quality, healthy, sustainability or size perceptions? Does this affect behavioural outcomes?
FR 33	Media	Attention; Cognitive reactions; Affective reactions; Motivational reactions	Do cartoons, drawings or photographs influence attention, cognitive, affective and motivational reactions in a different way? When does this occur? Does this affect food perceptions, attitudes and behavioural outcomes?
FR 34	Food porn	Food perceptions Attitude	What is the effect of food porn used at the point of purchase on product perceptions and attitudes? Does this affect behavioural outcomes?
FR 35	Food porn	Attention; Cognitive reactions; Affective reactions; Motivational reactions	Does food porn influence attention, affective, cognitive and motivational reactions? When does this occur? Does this explain effects on food perceptions, attitudes and behavioural outcomes?
**MATERIALITY**			
FR 36	Visual Texture	Food perceptions	Which visual texture factors affect quality, health, and sustainability, size perceptions and attitudes? Why and when does this occur? Does this affect behavioural outcomes?
FR 37	Visual Texture	Taste perception	Which taste inferences do people make based on the visual texture of the package or serving plate? When do these effects occur? Does this affect behavioural outcomes?
FR 38	Visual Texture	Attention; Cognitive reactions; Affective reactions; Motivational reactions	What is the effect of visual texture on attention, cognitive, affective and motivational reactions and how can they explain effects on food perceptions, attitudes and behavioural outcomes?
FR 39	Reflectance	Quality perception	What is the effect of glossy vs. matte packaging on food quality? Does this depend on the price of the food? Does this affect behavioural outcomes?
FR 40	Reflectance	Taste perception; Health perception	Do matte packaging signal healthier products whereas glossy packaging signal tastier products? Or vice versa? Why? Does this affect behavioural outcomes?
FR 41	Reflectance	Food perceptions; Attitude	What is the moderating role of food category on the effects of glossy (vs. matte) packaging on food perceptions, attitudes and behavioural outcomes?
FR 42	Reflectance	Attention; Cognitive reactions; Affective reactions; Motivational reactions	What is the effect of reflectance on attention, cognitive, affective and motivational reactions and how can they explain effects on food perceptions, attitudes and behavioural outcomes?
FR 43	Opacity	Quality perception	What is the effect of transparent packaging on food quality perceptions? Why does this occur? Does this affect behavioural outcomes?
FR 44	Opacity	Quality perception	Is the effect of transparent packaging on food quality perceptions contingent on material type (e.g., glass vs. plastics)?
FR 45	Opacity	Sustainability perception; Size perception	What is the effect of transparent packaging on sustainability perceptions and size estimations? Does this affect behavioural outcomes?
FR 46	Opacity	Food perceptions; Attitude	Where should a transparent window be placed to optimally affect food perceptions, attitudes and behavioural outcomes?
FR 47	Opacity	Attention; Cognitive reactions; Affective reactions; Motivational reactions	What is the effect of transparency on attention, cognitive, affective and motivational reactions and how can this explain product perceptions, attitudes and behavioural outcomes?
FR 48	Fluorescence	Food perceptions; Attitude; Attention; Cognitive reactions; Affective reactions; Motivational reactions	What is the effect of fluorescence of food, packaging, in-store or online material on food perceptions, attitudes, attention, cognitive, affective and motivational reactions? Does this affect behavioural outcomes?
**TEXT AND PICTURE COMBINATIONS**			
FR 49	Text-Picture combinations	Attitude	What is ideal combination of informational cues/transformational cues or text/picture in the context of healthy food packaging and how can this impact people’s healthy choices? What is the moderating effect of food involvement and type of product?
FR 50	Text-Picture combinations	Attention; Cognitive reactions; Affective reactions; Motivational reactions	What is the effect of specific word-picture ratio’s on attention, cognitive, affective and motivational reactions and how can they explain effects on food perceptions, attitudes and behavioural outcomes? Does this depend on the food category?
**LOCATION**			
FR 51	Verticality	Food perceptions; Attitude	What is the effect of vertical positioning of labels/verbal claims on a package on food perceptions, attitudes and behavioural outcomes. What is the effect of food type?
FR 52	Verticality	Attention; Cognitive reactions; Affective reactions; Motivational reactions	What is the effect of verticality of package labeling on attention, cognitive, affective and motivational reactions and how can they explain effects on food perceptions, attitudes and behavioural outcomes? Does this depend on the content of the label or the food category?
FR 53	Verticality	Attention; Cognitive reactions; Affective reactions; Motivational reactions	What is the effect of verticality of nutritional cues within a nutritional label on attention, cognitive, affective and motivational reactions and how can they explain effects on food perceptions, attitudes and behavioural outcomes? Does this depend on the food category?
FR 54	Horizontality	Food perceptions Attitude	What is the effect of horizontal positioning of labels on packaging on product perceptions and attitudes? Does this affect behavioural outcomes?
FR 55	Horizontality	Attention	What is the effect of horizontal positioning of labels on packaging on attention and how can they explain effects on food perceptions, attitudes and behavioural outcomes? Does this depend on label content?
FR 56	Direction	Cognitive reactions	Is package or product orientation processed before meaning of the product? Does this depend on the type of food?
FR 57	Direction	Food perceptions; Attitude	What is the effect of the vertical orientation of the display background on food perceptions of sustainable food products? Does this affect behavioural outcomes?
FR 58	Direction	Attitude	Would matching orientations of package handles with the right vs. left handedness of the person be equally important in case the person likes or dislikes the food product? Does this affect behavioural outcomes?
FR 59	Camera angle	Food perceptions Attitude	What is the effect of upward vs. downward vs. diners’ eye camera angle on (healthy and sustainable) food perceptions, attitudes and behavioural outcomes?
FR 60	Camera angle	Attention; Cognitive reactions; Affective reactions; Motivational reactions	What is the effect of upward vs. downward vs. diners’ eye camera angle, mobile device tilt angle or vertical vs. horizontal online shop orientation on attention, cognitive, affective and motivational reactions? Does this depend on food category? Does this affect behavioural outcomes?
FR 61	Spacing	Food perceptions; Attitude	What is the effect of spacing between package elements on food perceptions, attitudes and behavioural outcomes?
FR 62	Spacing	Attention; Cognitive reactions; Affective reactions; Motivational reactions	What is the effect of spacing between package elements on attention, cognitive, affective and motivational reactions? Does this depend on food category? Does this affect behavioural outcomes?
**MOVEMENT**			
FR 63	Movement	Food perceptions; Attitude	What is the effect of food product movement on food perceptions, attitudes and behavioural outcomes?
FR 64	Movement	Food perceptions; Attitude	What is the effect of Graphic Interchange Formats (GIFs) vs. static images of food on food perceptions, attitudes and behavioural outcomes? Does it depend on type of food? Does it depend on the speed of the movement of the GIF?
FR 65	Movement	Attention; Cognitive reactions; Affective reactions; Motivational reactions	What is the effect of approaching vs. receding actual or implied movements of food items on attention, cognitive, affective and motivational reactions? Does this depend on food category? Does it depend on the speed of moving? Does this affect behavioural outcomes?
FR 66	Movement	Attention; Cognitive reactions; Affective reactions; Motivational reactions	What is the effect of GIFs of food items on attention, cognitive, affective and motivational reactions? Does this depend on food category? Does it depend on the speed of moving? Does this affect behavioural outcomes?
**SPATIAL RELATION BETWEEN OBJECT AND SELF**			
FR 67	Spatial relation between object and self	Food perceptions; Attitude	What is the effect of first vs. third person perspective on food perceptions, attitudes and behavioural outcomes?
FR 68	Spatial relation between object and self	Attention; Cognitive reactions; Affective reactions; Motivational reactions	What is the effect of first vs. third person perspective on attention, cognitive, affective and motivational reactions? Does this affect behavioural outcomes?
**FURTHER CONSIDERATIONS**			
	Topic		Future Research Questions
FR 69	Combination of visual design cues		Which combinations of visual design cues positively affect behavioural outcomes? Is this contingent on food type? Which psychological processes can explain these effects?
FR 70	Cross model correspondence		What are moderators of cross modal correspondence?
FR 71	Online versus offline food contexts		Do the effects of visual design cues on behavioural outcomes differ in an online or offline food choice context? How can these different effects be explained?
FR 72	Mobile applications		How do visual design cues affect behavioural outcomes in mobile applications? How can these effects be explained? What are moderating factors?
FR 73	Long-term effects for unappealing products		Which visual design cues can be used to increase acceptance of unappealing products in the long run?
FR 74	Implicit and explicit measures		Are both implicit and explicit measures of psychological processes and behavioural outcomes necessary to optimally understand behavioural outcomes in a food choice context?

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
