# Peer review of "Visual Design Cues Impacting Food Choice: A Review and Future Research Agenda"

_foods, 2020, doi:10.3390/foods9101495_

Round 1

Reviewer 1 Report

Thank you for taking my comments into consideration and introducing them in the new version which seems clearer now.

The paper's contribution to the field is good. The paper's probability to stimulate future research is unquestionably high.

I still think that the length of the paper is EXTREMELY long as well as the number of references and it is advisable that this should be shortened.

Please, see my comments attached.

Author Response

Reply to the reviewers – Submission Foods-960684

We thank the editor, AE and reviewers for the interesting comments. We are pleased to read that the reviewers believe that the paper stimulates future research to an unquestionably high extent as this is one of our main objectives. It was thought-provoking reading alternative views on our framework and suggestions for optimization. Below we elaborate, point-by-point on the major comments and indicate how we tackled them.

Length of the paper

We followed the suggestions of both reviewers to shorten the length of the paper. As suggested by reviewer 2, we decided to indeed focus on in-store visual design cues that are managed by food producers (e.g., food packaging, food advertising, food displays). Visual design cues managed by retailers (e.g., shelf-positioning, food-positioning) are left out in order to reduce the length of the paper.

Effects of visual design cues in non-food contexts

Reviewer 2 suggests to provide a stronger argument on what is specific about visual design cues impacting food choice and not other (non-food) product choices. We want to clarify that we are not arguing that visual design cues that impact food choice do not impact choice of other consumer goods. We only focus on how they impact food choice given the scope of the journal Foods is about food. In addition, via creating multiple future research (FR) suggestions we reason that it would be interesting to investigate whether research from the non-food literature could be transferable to the food literature. We believe that the questions (FR’s) whether particular theories from non-food are transferable to food context is important since it is common practice that general theories are often further investigated and applied in different domains (e.g. Construal Level Theory is applied in retail settings by Van Kerckhove, Geuens and Vermeir, 2015 as well as in advertising domains: Roose, Vermeir, Geuens and Van Kerckhove, 2019). Cross-disciplinary research is as such also important for the research domain of food. By questioning whether general theories such as Construal Level Theory and Regulatory Fit theory are also applicable in food contexts, we aim to stretch that further research is required on this transferability. We do not aim or claim to have the answers.  By merely posing these FR questions we hope to expand the domain of food research.

We follow the reviewer that we wrongly formulated in our last reply to the reviewers that ‘a lot of the links that we suggest to investigate would also apply to a non-food context’. It is not our aim to make any suggestions beyond ‘food’ in a paper targeted at Foods, because we value and want to respect the specific focus of the journal. What we meant was that a lot of the links (FR) that we suggest to investigate are derived from and inspired by literature from non-food contexts.

Next, we added a reference (reference [34]) for the statement added in the previous review round: “Especially in a food choice context, choices are often made quite automatically without much thought. Often, people make choices based on visual design cues that attract their attention in the choice context.” (line: 53-55). We meant by this statement that food choices are often made without much thought and therefore they are easily influenced by contextual cues. We changed this to “In in a food choice context, choices are often made quite automatically without much thought based on cues that attract attention in the choice context [34]”.

[34] Cohen D.A., Babey S.H. Contextual influences on eating behaviours: Heuristic processing and dietary choices. Obes. Rev. 2012, 13, 766–779.

References

In the current version we order all references according to their chronological appearance in the paper. By reducing the length of the paper also the reference list is reduced.

Figure 1

We adapted “Figure 1: Overview of Visual Design cues, Psychological Processes and Behavioural Outcomes discussed in this paper” to make it more comprehensive: 1) All visual design categories and their subcomponents are mentioned in the figure.  2) Following Adaval et al. (2018) and Sample et al. (2020), we added a clear categorization of which cues are defined as ‘object processed cues’ and which cues are defined as ‘spatial processed cues.’3) Following Labrecque et al. (2013), we visualized how all visual design cues can have a direct impact on behavioural outcomes (see two arrows at the top and bottom of the figure). 4) We indicated how the visual design cues can have an impact on behavioural outcomes via all different types of psychological variables and a combination thereof. We intentionally kept the categorization of the psychological processes at a first level. Further specifying the subcomponents of the psychological process categories would not make sense given the sometimes unique relations of the specific visual design cue and the subcomponents of the psychological processes. 5) We indicated how all psychological process variable can influence the other psychological process variables (*). 6) In the title of Figure1 we refer to Adaval et al. (2018), Sample et al. (2020) and Labreqcue et al. (2013) as we used these papers as inspirational sources for our conceptual framework. Thanks to these adaptations we believe that this figure 1) shows the complete framework and 2) is indicative for our inspirational framework.

Headings

For each of the repetitive titles we included the visual design category name in the title. This change makes that all our titles are unique. An example of these changes is: previous title 4.2.1.1. “product perceptions and attitudes” is transformed into “4.2.1.1. Effect of Dimensionality of shape on product perceptions and attitudes.” We hope these changes 1) clarify the structure of the paper, 2) facilitates the reader to navigate through our paper and 3) helps the reader to quickly identify his/her exact location in the paper.

We would like to end this reply with thanking the reviewers. We believe that due to their constructive feedback the paper became a comprehensive overview of 1) visual design cues in in-store food contexts, 2) their (in)direct effects and 3) suggestions of future research based on the detected the literature gaps. Throughout the review process our scope become more narrow, but in the end this optimized the readability and comprehensibility of the paper.

With kind regards,

Iris Vermeir and Gudrun Roose

Reviewer 2 Report

The authors did not incorporate my main suggestions without providing a substantiated argument for doing so. Therefore my former comments still stand.

The authors point out in their revision notes that they "Furthermore, we think that a lot of the links that we suggest to investigate would also apply to a non-food context. We elaborated on that in the introduction.". It's not clear to me where and how this is done. I can see that there is an additional sentence, which might argue for the specific role of food product choices, but it is a strong claim without any citation. That is not acceptable:

  1. 63: "Especially in a food choice context, choices are often made quite automatically without 63 much thought. Often, people make choices based on visual design cues that attract their attention in 64 the choice context."

Concerning my second point on tightening the review to only focus on package design elements, the authors argue that in their opinion, the literature would be too scattered. In that we stand opinion against opinion. My argument is that shelf positioning and variety are not visual design elements, but belong to the domain of in-store marketing. They are managed by the retailers and not the producers and in that way relevant to a different readership. Having said that, for "variety" to count as a design cue is a bit of a stretch. Variety provides additional choice to the consumer to satisfy different preferences, it's not in that sense a design issue.

Author Response

Reply to the reviewers – Submission Foods-960684

We thank the editor, AE and reviewers for the interesting comments. We are pleased to read that the reviewers believe that the paper stimulates future research to an unquestionably high extent as this is one of our main objectives. It was thought-provoking reading alternative views on our framework and suggestions for optimization. Below we elaborate, point-by-point on the major comments and indicate how we tackled them.

Length of the paper

We followed the suggestions of both reviewers to shorten the length of the paper. As suggested by reviewer 2, we decided to indeed focus on in-store visual design cues that are managed by food producers (e.g., food packaging, food advertising, food displays). Visual design cues managed by retailers (e.g., shelf-positioning, food-positioning) are left out in order to reduce the length of the paper.

Effects of visual design cues in non-food contexts

Reviewer 2 suggests to provide a stronger argument on what is specific about visual design cues impacting food choice and not other (non-food) product choices. We want to clarify that we are not arguing that visual design cues that impact food choice do not impact choice of other consumer goods. We only focus on how they impact food choice given the scope of the journal Foods is about food. In addition, via creating multiple future research (FR) suggestions we reason that it would be interesting to investigate whether research from the non-food literature could be transferable to the food literature. We believe that the questions (FR’s) whether particular theories from non-food are transferable to food context is important since it is common practice that general theories are often further investigated and applied in different domains (e.g. Construal Level Theory is applied in retail settings by Van Kerckhove, Geuens and Vermeir, 2015 as well as in advertising domains: Roose, Vermeir, Geuens and Van Kerckhove, 2019). Cross-disciplinary research is as such also important for the research domain of food. By questioning whether general theories such as Construal Level Theory and Regulatory Fit theory are also applicable in food contexts, we aim to stretch that further research is required on this transferability. We do not aim or claim to have the answers.  By merely posing these FR questions we hope to expand the domain of food research.

We follow the reviewer that we wrongly formulated in our last reply to the reviewers that ‘a lot of the links that we suggest to investigate would also apply to a non-food context’. It is not our aim to make any suggestions beyond ‘food’ in a paper targeted at Foods, because we value and want to respect the specific focus of the journal. What we meant was that a lot of the links (FR) that we suggest to investigate are derived from and inspired by literature from non-food contexts.

Next, we added a reference (reference [34]) for the statement added in the previous review round: “Especially in a food choice context, choices are often made quite automatically without much thought. Often, people make choices based on visual design cues that attract their attention in the choice context.” (line: 53-55). We meant by this statement that food choices are often made without much thought and therefore they are easily influenced by contextual cues. We changed this to “In in a food choice context, choices are often made quite automatically without much thought based on cues that attract attention in the choice context [34]”.

[34] Cohen D.A., Babey S.H. Contextual influences on eating behaviours: Heuristic processing and dietary choices. Obes. Rev. 2012, 13, 766–779.

References

In the current version we order all references according to their chronological appearance in the paper. By reducing the length of the paper also the reference list is reduced.

Figure 1

We adapted “Figure 1: Overview of Visual Design cues, Psychological Processes and Behavioural Outcomes discussed in this paper” to make it more comprehensive: 1) All visual design categories and their subcomponents are mentioned in the figure.  2) Following Adaval et al. (2018) and Sample et al. (2020), we added a clear categorization of which cues are defined as ‘object processed cues’ and which cues are defined as ‘spatial processed cues.’3) Following Labrecque et al. (2013), we visualized how all visual design cues can have a direct impact on behavioural outcomes (see two arrows at the top and bottom of the figure). 4) We indicated how the visual design cues can have an impact on behavioural outcomes via all different types of psychological variables and a combination thereof. We intentionally kept the categorization of the psychological processes at a first level. Further specifying the subcomponents of the psychological process categories would not make sense given the sometimes unique relations of the specific visual design cue and the subcomponents of the psychological processes. 5) We indicated how all psychological process variable can influence the other psychological process variables (*). 6) In the title of Figure1 we refer to Adaval et al. (2018), Sample et al. (2020) and Labreqcue et al. (2013) as we used these papers as inspirational sources for our conceptual framework. Thanks to these adaptations we believe that this figure 1) shows the complete framework and 2) is indicative for our inspirational framework.

Headings

For each of the repetitive titles we included the visual design category name in the title. This change makes that all our titles are unique. An example of these changes is: previous title 4.2.1.1. “product perceptions and attitudes” is transformed into “4.2.1.1. Effect of Dimensionality of shape on product perceptions and attitudes.” We hope these changes 1) clarify the structure of the paper, 2) facilitates the reader to navigate through our paper and 3) helps the reader to quickly identify his/her exact location in the paper.

We would like to end this reply with thanking the reviewers. We believe that due to their constructive feedback the paper became a comprehensive overview of 1) visual design cues in in-store food contexts, 2) their (in)direct effects and 3) suggestions of future research based on the detected the literature gaps. Throughout the review process our scope become more narrow, but in the end this optimized the readability and comprehensibility of the paper.

With kind regards,

Iris Vermeir and Gudrun Roose

This manuscript is a resubmission of an earlier submission. The following is a list of the peer review reports and author responses from that submission.

Round 1

Reviewer 1 Report

See file attached

Reviewer 2 Report

Dear Authors,

I appreciated your manuscript “Visual design cues impacting food choice: a review and future research agenda” and I think that your study is a huge work in term of collection and analysis of papers dedicated to this topic. However, your contribution is a review, as you declare, and therefore it should be written following the review guidelines suggested by MDPI i.e. “Reviews: These provide concise and precise updates on the latest progress made in a given area of research. Systematic reviews should follow the PRISMA guidelines.”

In my opinion, your manuscript in the current view is not “concise” and hence it should be rewritten in order to provide in synthesis, the latest progress made in the selected research area that you defined.

I suggest to re-submit this review after some modifications, in line with review guidelines.

Best regards

Reviewer 3 Report

The author's of the manuscript "Visual design cues impacting food choice: a review and future research agenda" provide an extensive and interesting review of the current body of research on this topic. It is well structured and interesting to read.

I have two suggestions and some minor language issues.

My first suggestion regards the positioning of this manuscript. Of course it deals with food products, as it is under review at the journal foods. However, the author's could provide a stronger argument on what is specific about visual design cues impacting food choice and not other product choice. Would visual design cues that impact food choice NOT impact choice of other consumer goods? In some cases the authors refer to non-food literature to provide an outlook for further research, but the reader is left to whether the current findings are transferable or not. The authors might not be able to answer this question, but they could make a better case for their focus on food products - either in the introduction or in the discussion.

My second suggestion refers to the content of including some visual design elements, which are not packaging related (e.g. shelf layout), but not others (e.g. advertisements). The literature review here is extensive and for the most part deals with packaging. In order to reduce the amount of information for the reader and to streamline the presented research, I would advise to solely focus on visual design cues on packaging. Shelf layout belongs to the sphere of in-store marketing, which can include a whole range of other visual design factors (floor design, end-of-aisle displays, lighting,...) where the here presented results would have a better fit.

Minor issues:

l.45 "at the mean time" - "at the same time"? "in the mean time"?

Figure 1  - contains some typos and is a bit blurry. That should be improved

Table 1 and 2 - should be on one page each to be understandable. Right now they seem quite messy as it is not clear why some of the columns to the lower end of the tables remain empty.

l.441 "higher food perception" - not sure what higher perception should mean

l.783 - "did not found" should be "did not find"

l.830 - "people'" - s missing

l.842,846 - "arts" - not sure what that means here

l.844 - "considerably" should be "considerable"

l.865 "Orth et al" has smaller font than the rest

l.903 "unsymmetric" - "asymmetric" is more common to use

ll.1133-1153 has larger font than the rest

l.1248 "sotres" - stores?

l.1319 first explain "construal level theory" here, even though it is used earlier in the manuscript. Should be defined at first usage

ll.1375-1384 has larger font than the rest

l.1742 - no sure people "suffer" from food neophobia. It's not a disease.

Thank you for putting so much work into synthesizing this stream of literature.